# Testing sensory evidence against mnemonic templates

**Nicholas E Myers[1,2]\*, Gustavo Rohenkohl[3], Valentin Wyart[4], Mark W Woolrich[2,5], Anna C Nobre[1,2], Mark G Stokes[1,2]\***

[1]Department of Experimental Psychology, University of Oxford, Oxford, United Kingdom; [2]Oxford Centre for Human Brain Activity, University of Oxford, Oxford, United Kingdom; [3]Ernst Strüngmann Institute for Neuroscience, Frankfurt, Germany; [4]Laboratoire de Neurosciences Cognitives, Département d'Etudes Cognitives, Ecole Normale Supérieure, Paris, France; [5]Oxford Centre for Functional MRI of the Brain, University of Oxford, Oxford, United Kingdom

**Abstract** Most perceptual decisions require comparisons between current input and an internal template. Classic studies propose that templates are encoded in sustained activity of sensory neurons. However, stimulus encoding is itself dynamic, tracing a complex trajectory through activity space. Which part of this trajectory is pre-activated to reflect the template? Here we recorded magneto- and electroencephalography during a visual target-detection task, and used pattern analyses to decode template, stimulus, and decision-variable representation. Our findings ran counter to the dominant model of sustained pre-activation. Instead, template information emerged transiently around stimulus onset and quickly subsided. Cross-generalization between stimulus and template coding, indicating a shared neural representation, occurred only briefly. Our results are compatible with the proposal that template representation relies on a matched filter, transforming input into task-appropriate output. This proposal was consistent with a signed difference response at the perceptual decision stage, which can be explained by a simple neural model.

**\*For correspondence:** nicholas. myers@ohba.ox.ac.uk (NEM); mark.stokes@ohba.ox.ac.uk (MGS)

**Competing interests:** The authors declare that no competing interests exist.

## Introduction

Human perception is flexible: the dimensions guiding perceptual decisions can be updated rapidly as a function of the current task. When decisions are based on perceptual analysis, task goals influence behaviour by creating an internal template: incoming sensory information is then matched against it. The representation of templates therefore plays a fundamental role in guiding perception and decision-making. Biased competition (*Desimone and Duncan, 1995*) provides a broad framework for how the brain interprets new sensory information in light of the current search template. A central tenet is that attention tonically pre-activates visual cortical neurons with receptive fields for relevant, template-matching stimuli (*Reynolds and Chelazzi, 2004*; *Chelazzi et al., 2011*). Single-cell neurophysiology (*Chelazzi et al., 1993*; *Luck et al., 1997*; *Chelazzi et al., 1998*) and human functional magnetic resonance imaging (fMRI; *Chawla et al., 1999*; *Kastner and Ungerleider, 2000*; *Silver et al., 2007*; *Kastner et al., 2009*; *Reddy et al., 2009*) have demonstrated that template representation and stimulus processing can occur in overlapping neural populations in the visual cortex. Moreover, stimulus and template activity patterns cross-generalize (when measured with fMRI, *Stokes et al., 2009*), implying that the two share a common neural code. In the simplest case, increasing baseline activity of a stimulus-specific representation could boost target processing (*Sylvester et al., 2009*). This boost could facilitate target selection and reduce distractor competition for downstream processing resources (*Bundesen et al., 2005*; *Maunsell and Treue, 2006*).

**eLife digest** Imagine searching for your house keys on a cluttered desk. Your eyes scan different items until they eventually find the keys you are looking for. How the brain represents an internal template of the target of your search (the keys, in this example) has been a much-debated topic in neuroscience for the past 30 years. Previous research has indicated that neurons specialized for detecting the sought-after object when it is in view are also pre-activated when we are seeking it. This would mean that these 'template' neurons are active the entire time that we are searching.

*Myers et al.* recorded brain activity from human volunteers using a non-invasive technique called magnetoencephalography (MEG) as they tried to detect when a particular shape appeared on a computer screen. The patterns of brain activity could be analyzed to identify the template that observers had in mind, and to trace when it became active. This revealed that the template was only activated around the time when a target was likely to appear, after which the activation pattern quickly subsided again.

*Myers et al.* also found that holding a template in mind largely activated different groups of neurons to those activated when seeing the same shape appear on a computer screen. This is contrary to the idea that the same cells are responsible both for maintaining a template and for perceiving its presence in our surroundings.

The brief activation of the template suggests that templates may come online mainly to filter new sensory evidence to detect targets. This mechanism could be advantageous because it lowers the amount of neural activity (and hence energy) needed for the task. Although this points to a more efficient way in which the brain searches for targets, these findings need to be replicated using other methods and task settings to confirm whether the brain generally uses templates in this way.

However, recent findings complicate this simple model. Population-level analyses of time-resolved neural recordings show that stimulus decoding is highly time-specific (*King and Dehaene, 2014*), with discriminative activity patterns changing at the millisecond scale. Such dynamic coding has been observed at the level of population spiking patterns within individual brain areas (*Meyers et al., 2008*; *Crowe et al., 2010*; *Stokes et al., 2013*), and at the level of distributed activation patterns across the cortex (*King et al., 2013*; *Cichy et al., 2014*; *Wolff et al., 2015*), suggesting that this temporal dimension is an inherent aspect of neural coding (*Buonomano and Maass, 2009*). Importantly, neural populations in visual (*Meyers et al., 2008*; *Sreenivasan et al., 2014*) and prefrontal cortex (*Hussar and Pasternak, 2012*; *2013*; *Stokes et al., 2013*; *Astrand et al., 2015*) appear to represent a memorized stimulus with an independent pattern from that used during initial encoding. As a consequence, it is necessary to distinguish between a neural *pattern* (which may vary from moment to moment), and the *representational content* that is encoded in that pattern (which may be stable even when the pattern changes over time, see *Haxby et al., 2014*).

The highly dynamic trajectory that stimulus processing traces through activation state-space challenges classic models of template representation. These propose tonic activation of a static neural pattern, begging the question: which of the many points along the processing trajectory should be pre-activated?

An alternative scheme enables templates to guide perceptual decision-making even when stimulus processing is dynamic. If stimulus and template representations rely on different patterns of neural activity in the circuit, then a matched-filter process (c.f. *Sugase-Miyamoto et al., 2008*; *Nikolic et al., 2009*; *Stokes, 2015*) could be envisaged in which the dynamic pattern of stimulus encoding would be automatically transformed into a pattern reflecting the degree of overlap to the template. This could be achieved if the pattern of activity elicited by the incoming stimulus is weighted by the neural pattern associated with the stored template information.

While visual templates for target detection have been central to attention research, their role has been somewhat neglected in the study of perceptual decision-making. Perceptual decision-making tasks usually require the judgment of a visual stimulus feature against a fixed decision boundary or template (*Gold and Shadlen, 2007*). These tasks typically require judgments to be made at varying levels of perceptual difficulty (*Vogels and Orban, 1990*; *Ghose et al., 2002*; *Purushothaman and*

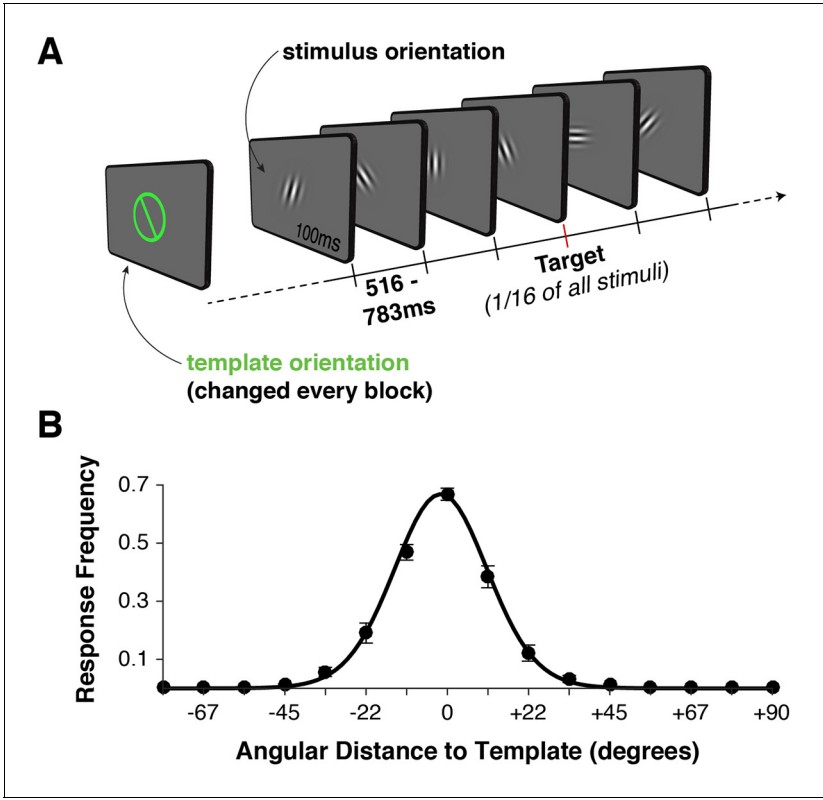

**Figure 1.** Task design and behavior. (**A**) Each block began with the presentation of a target orientation, which observers maintained for the duration of a task block. Template presentation was followed by a serial stream of randomly oriented stimuli. Observers were asked to respond with a button press whenever the stimulus matched the template orientation. (**B**) Response frequency was highest for target trials, and dropped off sharply for non-targets with increasing angular distance between template and stimulus orientation. Error bars indicate standard error of the mean across observers. The black line denotes a von Mises distribution fit to the responses.

The following figure supplement is available for figure 1:

**Figure supplement 1.** Reaction Time Distribution and Effects of Target Proximity.

*Bradley, 2005*; *Summerfield and Koechlin, 2008*; *Scolari and Serences, 2010*; *Wyart et al, 2012*). The majority of perceptual decision-making studies have kept the decision boundary (or template) constant over the entire experiment, impeding a clear evaluation of the representation of templates as distinct from stimulus representation and from the sensory-to-template comparison. In the present study, we varied template and stimulus values independently, enabling us to examine the extent to which their coding and their temporal profiles overlap. We used pattern analysis of simultaneously recorded magneto- (MEG) and electroencephalography (EEG) to track visual template matching with high temporal resolution as human observers performed a parametric match-to-template orientation task (*Figure 1A*).

Neural responses rapidly traversed a cascade of discriminative patterns, transforming the initial task-invariant stimulus code, in conjunction with the template code, into a decision-relevant code. Template patterns and stimulus patterns cross-generalized only in a short time window during initial processing, suggesting some independence in the two neural codes. Despite these differences in the neural patterns, the content of the representation encoded in these patterns (as measured by their representational similarity) corresponded over a more sustained period. This might be expected if templates are encoded as a matched filter in the connections between stimulus-sensitive and decision-relevant populations.

Interestingly, after the stimulus information was already reliably present and the response-relevant information had begun to emerge, neural signals also encoded the (task-irrelevant) *signed* difference between the current stimulus and the search template. This processing stage additionally

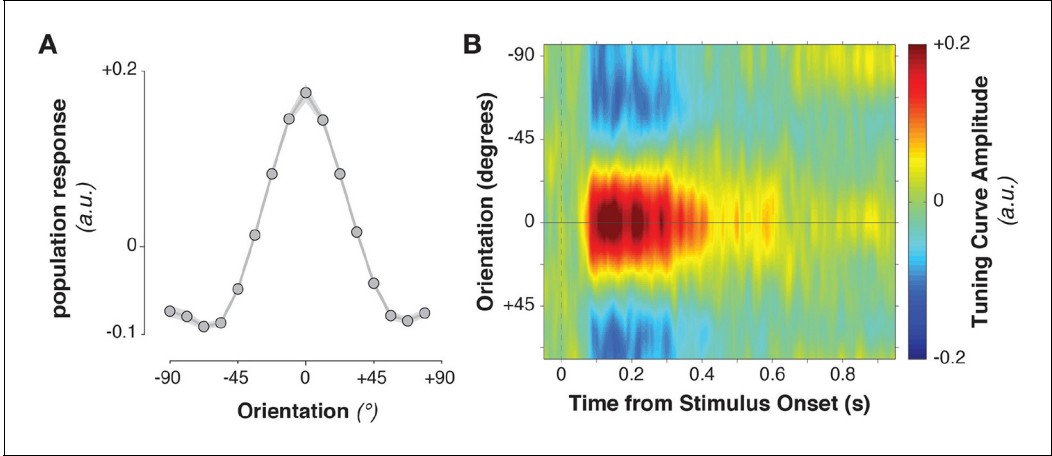

**Figure 2.** Stimulus-evoked population tuning curves. (**A**) Average population tuning curve, 50–300 ms after stimulus onset. (**B**) Time-resolved population tuning curve, showing a sharp increase in the tuning curve slope shortly after stimulus onset, tapering off within 500 ms.

suggests the presence of a matched filter that permits the flexible calculation of deviations from a search template. We argue, on the basis of a simple neural model, that this effect is consistent with the use of a population code for perceptual decision-making (*Ma et al., 2006*; *Zemel et al., 1998*; *Beck et al., 2008*).

## Results

### Behavior

We recorded simultaneous MEG and EEG signals from 10 human observers as they performed a serial visual match-to-template task (see *Figure 1A* and Materials and methods). At the beginning of each block, observers viewed a target orientation to be maintained in memory and used as a search template for the duration of the block. Each block consisted of a centrally presented stream of Gabor patches (randomly drawn from a distribution of 16 orientations, uniformly spaced along the circle). Observers were instructed to respond with a button press whenever the target appeared. Over two sessions, each observer viewed a total of 7680 stimuli to maximize the statistical power of within-participant pattern analyses. On average, observers correctly detected approximately 70% of targets (*Figure 1B*). They also made a large proportion of false alarms to near targets (approximately 50% for offsets from the target angle of ± 11.25°), with false alarms rapidly dropping for more distant non-targets. Reaction times were distributed around 550 ms (*Figure 1—figure supplement 1A*), with no strong effect of target proximity on reaction time (p > 0.35, *Figure 1—figure supplement 1B*).

### MEG/EEG signals reflect population tuning curves for stimulus and template orientations

The stimulus information encoded in MEG/EEG signals was captured by calculating time-resolved population tuning curves (*Figure 2*, see Materials and methods). This approach transforms sensor-level responses into responses of virtual stimulus orientation channels: if a stimulus orientation is reflected in the MEG/EEG signal, virtual channel responses should peak at the corresponding orientation. In order to calculate the transformation of sensor data to tuning curves, the data were split into training and test sets. The training data were used to calculate each sensor's sensitivity to each stimulus orientation, yielding a weight matrix. This weight matrix was then multiplied with the data in the independent test set and averaged over sensors. Single-trial virtual channel responses were then centered on the orientation presented on that trial and averaged over trials, providing an average population tuning curve. Tuning curves were calculated separately for each time point in the trial. Stimulus tuning curves showed that MEG/EEG signals (EEG sensors were added to the analysis

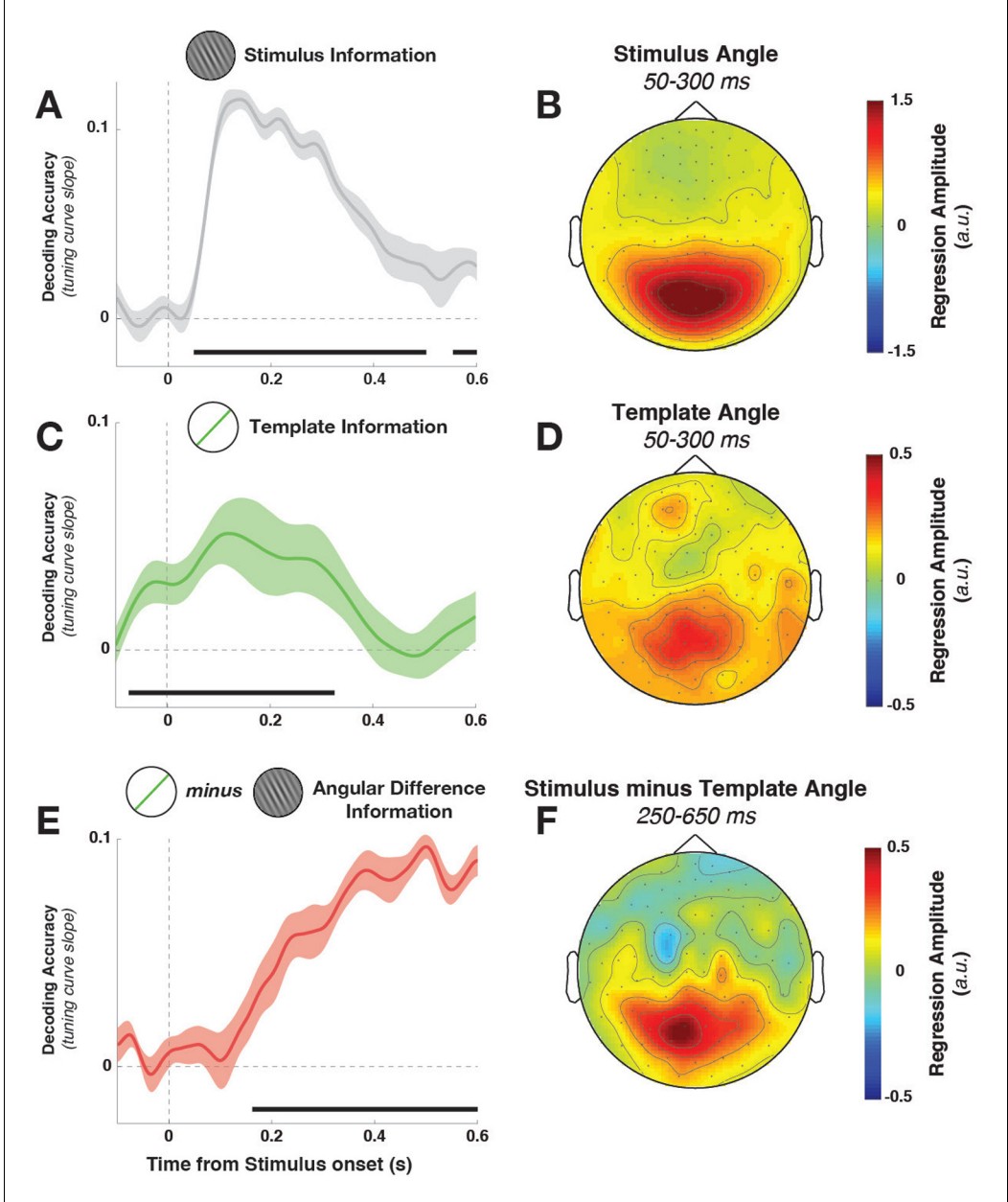

**Figure 3.** Task variable representation using population tuning curves (see *Figure 2*). (A) Stimulus orientation was represented in the early visual response. We fit weights (using linear regression of stimulus orientation on the neural response) using all trials in all training blocks and estimated virtual channel responses in the test block. Orientation-specific coding was estimated by calculating the linear slope of the tuning curve (between 0° and 90°). Consistent positive slopes indicate orientation selectivity at a given time point. Shading indicates between-subject standard error of the mean. Black bars denote significant time points (cluster-corrected). (B) Univariate sensitivity for stimulus orientation, calculated at each sensor and time point. Topography shows the shuffle-corrected orientation sensitivity (z-scored against a distribution generated from permuting stimulus orientations 1000 times), averaged across sensor triplets (two orthogonal planar gradiometers and one magnetometer) and across the stimulus-decoding window. Color coding denotes the z-score, averaged across observers. (C) Tuning curve slope and topography (D) for template orientation sensitivity. E and F show the same analyses, sorting trials by the angular distance between template and stimulus (i.e., the decision value).

in combined MEG/EEG sessions) reflected stimulus orientation shortly after its onset (*Figure 2B*, *Figure 3B,C*, *Figure 3A*, 52–500 ms relative to stimulus onset, cluster-corrected p = 0.0019).

Template orientation information was also present in the MEG/EEG signal during stimulus processing (*Figure 3C*, –72 to 324 ms, cluster p = 0.0045). A jack-knife analysis comparing onset

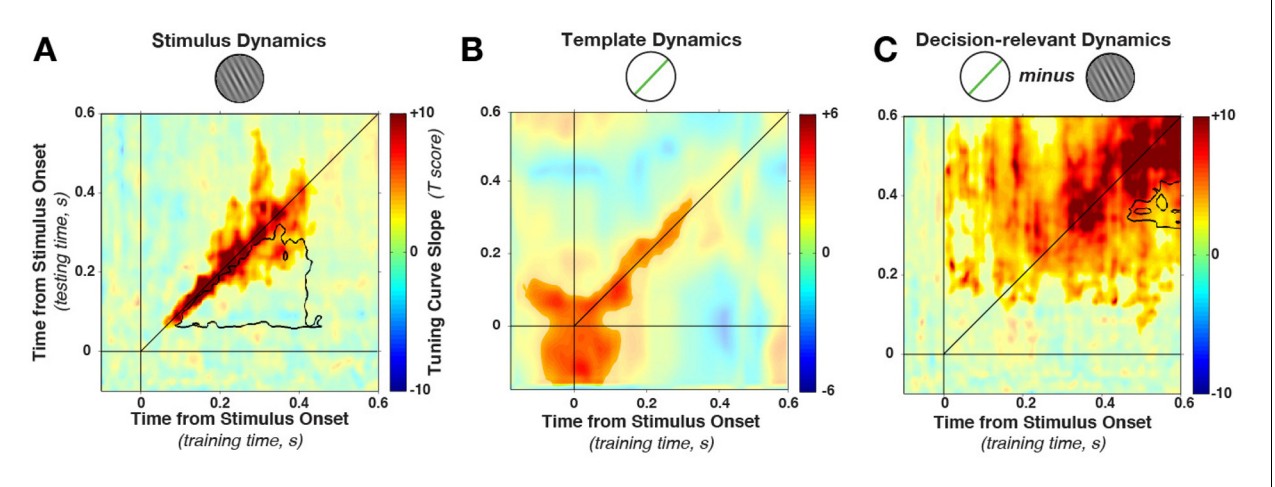

**Figure 4.** Cross-temporal generalization of orientation decoding. (**A**) Tuning curve amplitude for stimulus orientation, estimated by calculating weights at one time point and applying them to test data at all time points in a trial. While decoding is consistently high along the diagonal (in the time window that contains significant stimulus information, between 52 and 544 ms, significant cluster indicated by color saturation/opacity), the slope drops sharply at off-diagonal train-test time coordinates. This indicates that the discriminative patterns are not consistent across time—rather they change rapidly, even while the stimulus can be readily decoded (i.e. off-diagonal decoding is significantly lower than on-diagonal decoding, black outline). **B** and **C** show the same analyses as in A, but sorting all trials by the template angle and the decision-relevant angular distance, respectively.

latencies between template and stimulus coding showed that template coding began significantly earlier (template: $-72 \pm 13$ ms, stimulus: $52 \pm 6$ ms, $t_9 = -8.61$, $p = 3*10^{-6}$).

To track the temporal evolution of task-relevant coding (i.e., the decision-value), we also decoded the distance of the current stimulus to the current template (i.e. the signed angular distance between stimulus and template angles, from here on simply 'angular distance'). A strong effect of angular distance emerged around 200 ms (*Figure 3E*, 164–596 ms, cluster p = 0.0024), with an onset that was significantly later than for the stimulus orientation, $t_9 = 3.25$, $p = 0.002$). This effect was also present when training only on trials without a response (172–596 ms, cluster p = 0.0014), which discounts the possibility that this analysis simply reflected the difference in neural signals between responded trials (which were most frequent for small angular distances) and non-responded trials (most frequent for large angular distances).

While the three main task variables (stimulus, template, and angular distance) were all present in the signal, it is unclear whether different brain regions are involved. We calculated the sensor-level univariate sensitivity to each variable (at MEG sensors only, averaging across the magnetometer and two gradiometers at each location) to determine the topographies associated with different task variables. To a first approximation, all three variables were encoded in signals in visuo-parietal sensors (*Figure 3B,D,F*). While sensitivity was again strongest to stimulus orientation, template and angular-distance responses nonetheless showed very similar topographies, indicating that all three variables might be computed in overlapping or nearby populations.

## Stimulus and task activity patterns vary dynamically throughout the epoch

To examine whether patterns of stimulus activity changed dynamically throughout the epoch, we tested for cross-temporal generalization of decoding (as elaborated in *King and Dehaene, 2014*). The population tuning curve approach was extended across time by calculating weights on one time point in the trial (on a training data set) and applying those weights to all time points in the trial (on the left-out test data set).

Stimulus orientation decoding was significant in one main cluster along the diagonal (64–544 ms in training time, 52–436 ms in test time, cluster p = 0.0024, significant cluster extent indicated by color saturation in *Figure 4A*). More importantly, stimulus decoding was time-specific, with decoding dropping at off-diagonal train-test time points. To quantify the degree of dynamic coding

statistically, we evaluated the off-diagonal results using a conjunction t-test: each off-diagonal combination of timepoints ($t_{1,2}$) was compared against both on-diagonal within-time pairs ($t_{1,1}$ and $t_{2,2}$). Evidence for dynamic decoding was inferred if decoding for $t_{1,2}$ was significantly lower than both $t_{1,1}$ and $t_{2,2}$.

This drop-off is characteristic of dynamic coding (*King and Dehaene, 2014*; *Stokes, 2015*): despite significant decoding at two respective time points, the discriminative patterns do not generalize from one time point to the other. Off-diagonal generalization was significantly lower in a cluster (black outline in *Figure 4A*) stretching from 52–304 ms (training time) and from 88–436 ms (generalization time, cluster p = 0.0031, based on cluster extent, see Materials and methods). Since cross-generalization across time was significantly worse than within-time decoding, multiple stimulus-specific activity patterns seem to have been triggered in sequence.

Importantly, *Figure 4A* shows that the cluster of significant decoding (indicated by color saturation) and the cluster of significant dynamic coding (indicated by black outline) partially overlap. In this overlapping region, training on timepoint $t_1$ and testing on $t_2$ still leads to significant decoding, but this generalization across time is nonetheless significantly lower than training and testing at either $t_1$ or $t_2$ alone. Such overlap can occur if decoding draws on a combination of dynamic and stationary patterns during the same epoch (see also below). It is perhaps also interesting to note that we do not observe any evidence for periodic reactivation of orientation-specific patterns, which would be expected if the discriminating signal was oscillatory and phase-locked to the stimulus presentation (*King and Dehaene, 2014*).

Template information (*Figure 4B*) was present in an early cluster (training time: −140–340 ms, test time: −140–316 ms, relative to stimulus onset, cluster p = 0.0191). In contrast to the stimulus decoding, template decoding showed no significant attenuation of decoding on the off-diagonal.

Decision-relevant angular-distance decoding showed a dynamic pattern, although off-diagonal decoding appeared to be more pronounced (*Figure 4C*) compared with stimulus orientation decoding (*Figure 4A*). Nevertheless, the strongest decoding was along the diagonal (training time: 4–592 ms, test time: 64–592 ms, cluster p = 0.0009), with significantly reduced off-diagonal generalization in this time window (training time 316–424 ms, generalization time 472–592 ms, cluster p = 0.008). Since off-diagonal decoding was nonetheless significant in a large time window, it is possible that the angular distance exhibits both a dynamic and a static element. This could happen for two reasons. First, it could occur if one population follows a dynamic processing cascade, while a separate population is tonically active in response to a given angular distance. Additionally, significant off-diagonal generalization could occur if there is temporal variability in the underlying processes, smoothing out the dynamic time-specificity across trials.

## Cross-generalization between stimulus and template neural patterns

Training the population tuning-curve weights on template orientations around the time of stimulus onset (–150 to +300 ms) showed a strong trend toward generalizing to *stimulus* decoding shortly *after* onset (*Figure 5A*, 52–124 ms, corr. p = 0.063). Using only the pre-stimulus time window (–150 to 0 ms) to train the template pattern, stimulus information could still be extracted in this immediate post-stimulus period (*Figure 5B*, average over 50–150 ms after stimulus onset, $t_9 = 2.45$, p = 0.037), indicating that template-specific neural patterns may be pre-activated immediately before stimulus onset. This result indicates that template activity patterns and stimulus activity patterns do cross-generalize (e.g., *Stokes et al., 2009*), but only transiently. The template- and the stimulus-discriminative patterns correspond only in the earliest encoding period, but not later (even though stimulus decoding itself persisted up to 500 ms).

## Stable representational structure for stimulus and task variables

While the underlying patterns separating different stimulus orientations change dynamically after stimulus onset, the information content that is represented might be more stable. The basic decoding analysis already implies that dynamic neural patterns contain stable information: the same basis set is used for decoding throughout the epoch. Therefore, significant decoding along the time-specific diagonal axis in the cross-temporal analysis suggests that stimulus orientation (*Figure 4A*), or angular difference (*Figure 4C*), is represented throughout significant changes in the underlying neural patterns. However, a more formal test of the representational structure of multivariate activity is

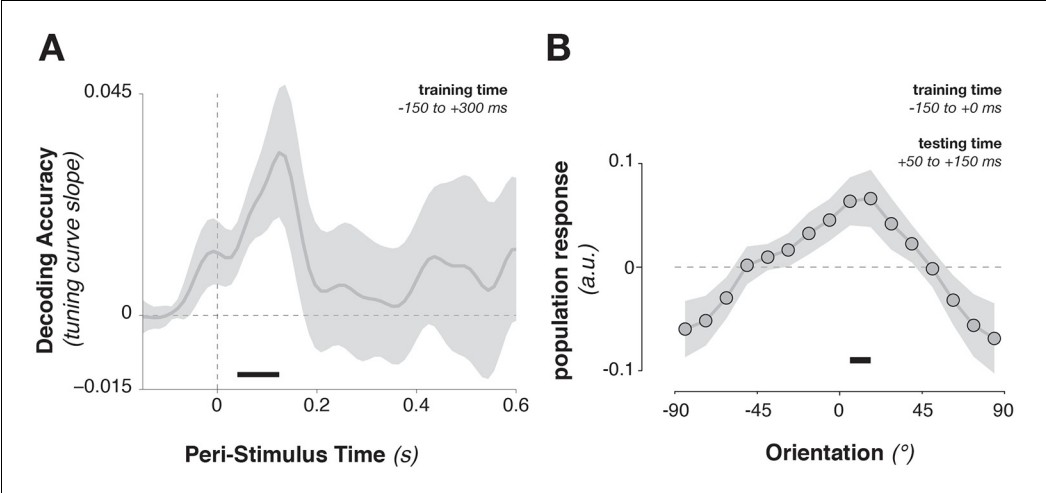

**Figure 5.** Cross-generalization from template-discriminative patterns to stimulus-discriminative patterns. (A) Calculating tuning-curve weights relative to the template orientations in a training data set (in window from –150 to +300 ms around stimulus onset), applying these weights on test data, and sorting them relative to the stimulus orientation, showed decoding early after stimulus onset that quickly returned to baseline. (B) Calculating population weights only on the pre-stimulus period (with respect to the template orientations) yielded a population tuning curve with a significant peak around the presented stimulus orientation (e.g. a significant peak above the average response around 0°, and a significant positive tuning curve slope between ± 90° and 0°). Shading indicates the standard error of the mean. Black bars indicate significant time points or orientations (p < 0.05).

provided by representational similarity analysis (RSA; *Kriegeskorte et al., 2008*). Rather than testing for discrimination per se, this approach focuses on the second-order pattern of condition-specific differences. This allows us to characterise the representational structure of the population code independently of the specific neural patterns associated with different stimulus orientations (*Kriegeskorte and Kievit, 2013*). As an example, a 40°-tilted stimulus might elicit a topography shortly after onset that is more similar to the topography from a 30°-stimulus than that of a 90°-stimulus. While the MEG patterns separating these stimuli might change throughout the trial, the relative difference between them could be preserved. This would indicate that the same kind of information about the stimuli is represented. This approach has been developed specifically to characterize implementation-independent representational geometry (*Kriegeskorte and Kievit, 2013*), and therefore is well-suited here to test whether dynamic neural patterns essentially code the same information.

We tested this by repeating the cross-temporal analyses on the dissimilarity relationships between MEG responses evoked by different orientations. Dissimilarity was quantified by the Mahalanobis distance matrices between all pairs of stimulus orientations (on one half of trials), separately for each time point in the trial. At each timepoint, this yielded a 16×16 distance matrix. We next calculated the same distance matrix for the remaining half of the data, and calculated the Pearson correlation coefficients between distance matrices from the two independent data sets, for each combination of time points.

In contrast to the dynamically varying stimulus-discriminative patterns, the representational similarity remained much more stationary (*Figure 6A*), with a stable plateau of high correlations (Fisher-transformed Pearson's rho) from the earliest time of stimulus decoding. We found a significant (on- and off-diagonal) cluster early in the epoch (28–596 ms, cluster p = 0.005). The temporal stability of the representation was, as above, tested by examining whether on-diagonal similarity was higher than off-diagonal similarity. While one short-lived dynamic cluster emerged (training time 82–146 ms, generalization time 130–226 ms, cluster p=0.0075, see black outline in *Figure 6A*), the majority of the epoch was dominated by time-stable correlations.

Similarly, the representational similarity of angular distance was stable throughout the trial (*Figure 6B*). Dissimilarity matrices correlated significantly in a later window in the trial (172–588 ms, cluster p = 0.0026), with no time points where within-time correlations were significantly higher than between-time correlations (all p > 0.20). These complementary analyses highlight the cardinal

feature of dynamic coding: discriminative dimensions vary with time even though the information content remains constant (*Laurent, 2002*; *Stokes, 2011*).

## Circular representational structure suggests a common coding scheme between stimuli and templates

As the previous section indicates, the representational similarity of different stimulus orientations is more temporally stable than the underlying discriminative pattern. If RSA can reveal stable representations over time, it could also uncover representational similarity between the template and the stimulus. In other words, even though the MEG patterns did not persistently cross-generalize between stimulus and template decoding, the dissimilarity matrices calculated for template orientations and for stimulus orientations might reveal a more stable match. This would indicate that similar content is being stored about stimuli and templates, even though the precise neural implementation might differ.

To quantify this relationship, we again tested for cross-temporal generalization of the dissimilarity matrix. However, here we correlated the dissimilarity matrix calculated between template orientations with the matrix calculated between stimulus orientations. Specifically, we correlated the Mahalanobis distance matrix between all eight template orientations, calculated at each time point, with the distance matrix between stimulus orientations (limiting our analyses to the same eight stimulus orientations that served as targets in the experimental session). The template-sorted dissimilarity matrix correlated significantly with the stimulus-sorted dissimilarity around the time of visual processing (cluster in *Figure 7A*, template structure from –48 to 196 ms, stimulus structure from 88–208 ms, cluster p = 0.038). The within-time comparison between templates and stimuli (i.e., the values along the diagonal) also showed a significant correlation (*Figure 7B*, 104–176 ms, p = 0.036, with trends toward significance between 412–464 ms, p = 0.079, and 552–596 ms, p = 0.086).

What is the basis of this representational similarity between templates and stimuli? Given the simplicity of the stimulus set, a straightforward representational structure comes to mind: more similar stimulus (or template) orientations evoke more similar MEG topographies. However, this cannot be deduced from the population tuning-curve analysis alone. To evaluate the possibility, we calculated neural dissimilarity matrices between the mean responses evoked by each of the 16 stimulus orientations, and projected this 16×16 neural dissimilarity matrix into two dimensions for visualization (using multi-dimensional scaling, *Figure 7D*). During the stimulus-encoding period (50–250 ms after stimulus onset), conditions fell onto a well-ordered circle: topographies were more similar (had a smaller Mahalanobis distance) if they were evoked by more similar stimulus orientations. This geometry was not present in the data before the onset of stimulus processing (–50 to +50 ms relative to stimulus onset, *Figure 7C*). We tested for the temporal stability of this representational structure by correlating the data-derived (Mahalanobis) distance matrix at each time point with an idealized distance matrix, derived from the angular distances of the respective stimulus orientations (i.e. a 16-point simplex, corresponding to the pairwise angular distances between all 16 presented orientations, *Figure 7F*). We used linear regression to fit the idealized distance matrix of the stimuli to the neural distance matrix at each time point. The stimulus similarity matrix significantly fit the neural data (*Figure 7E*, 44–432 ms, cluster p = 0.002).

Likewise, the neural dissimilarity matrix between different *template* orientations was well described by the same simplex structure (*Figure 7G,H*, 48–300 ms, cluster p = 0.0012, with a second cluster around the time of the next stimulus). Therefore, while the discriminative patterns for stimuli and templates cross-generalized only briefly, the content of their representation appears to be both stable over time and similar between task variables.

## MEG responses represent the task-irrelevant sign of the stimulus-template angular difference

We tested whether the decision-relevant angular distances (between the current stimulus and the template), which were necessary for guiding behavior, showed a comparable similarity structure. The angular distance could be decomposed into two components: its magnitude and its sign. The magnitude of the angular distance (i.e. the absolute difference between stimulus and template orientation) solely determined the task-relevance of a stimulus: the closer the magnitude is to 0°, the more likely the stimulus led to a target response. By contrast, the sign of the angular distance (i.e., whether a

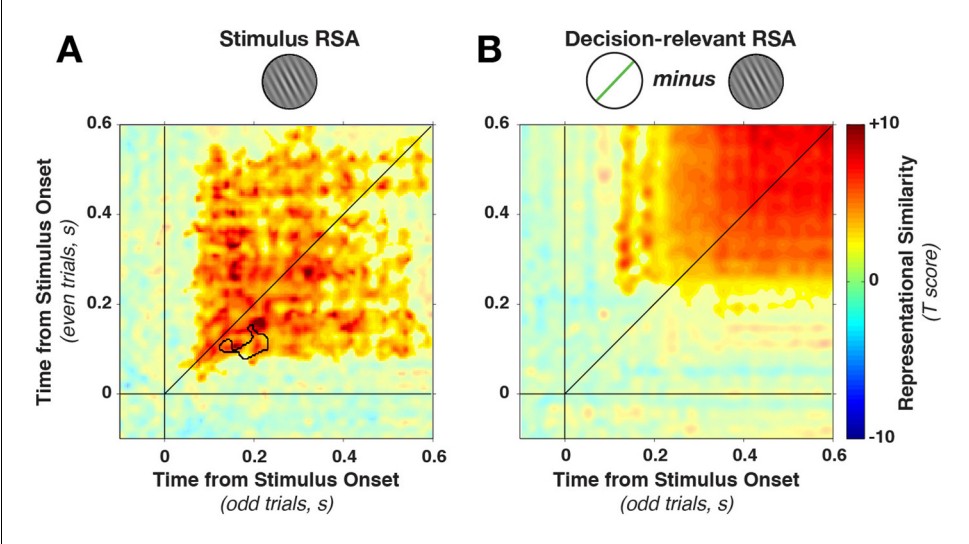

**Figure 6.** Cross-temporal generalization of representational similarity. (**A**) Pearson correlations between stimulus-orientation-sorted distance matrices, calculated at different time points and on independent data sets. Color saturation shows significant cluster at the group level (permutation test). The cluster extends off the diagonal in a square, indicating substantial cross-temporal generalization. In addition, there is a small dynamic cluster (black outline), meaning that pairs of time points within the black outline showed significantly lower correlations than their corresponding time points on the diagonal (even though they were still significantly greater than 0). (**B**) shows the same analysis as in A, but sorting all trials by the decision-relevant angular distance. There were no significant dynamic clusters. RSA, representational similarity analysis.

stimulus was oriented clockwise or counter-clockwise with respect to the template) had no relevance to the task, because it did not influence how close that stimulus was to the current template. In the next analysis, we therefore attempted to isolate the effects of magnitude and sign on the neural response. We projected (using independently calculated weights, see Materials and methods) neural responses (from all 16 possible angular distances, and from all MEG/EEG sensors) onto two axes measuring separately the influence of magnitude and sign of the angular distance at each point in the trial (*Mante et al., 2013*). This allowed an analysis of the MEG signal's sensitivity to the decision-relevant magnitude (measured by the amplitude of the response along the magnitude axis), independently of its sensitivity to the decision-irrelevant sign of the angular distance (measured along the sign axis).

The mean responses to the 16 angular distances fell roughly onto a circle that stretched out along the magnitude axis, with targets and near-targets clearly separable from the definite non-targets (*Figure 8A*). Interestingly, near non-targets that were either clockwise or counter-clockwise to the target also separated along the decision-irrelevant sign axis, indicating an unexpected result: angular distances with equal magnitude but different sign (i.e., stimuli at an identical distance to the template orientation, such as −23° and +23°) evoked distinct and separable neural responses. Mean projections along the task-irrelevant axis for conditions with equal magnitude but different sign diverged around the time of decision formation (348–588 ms, corrected p = 0.004, *Figure 8B*). We verified that this was the case even without relying on the task-projection approach by calculating Mahalanobis distances between pairs of angular distance trials that had equal magnitude (i.e., −11° vs. +11°, −22° vs. +22°, −34° vs. +34°), and found similar results (see *Figure 8—figure supplement 1*). In addition, we confirmed in a control analysis that different angular distances were not separable merely because of possible differences in response bias (*Figure 8—figure supplement 2*).

## Template matching based on probabilistic population codes

The neural encoding and sustained representation of the *signed* difference between the current stimulus and the template was unexpected because representing the sign of the angular distance was not necessary for solving the task (since it would be sufficient to calculate only the magnitude). However, the result yields some insight into the particular neural implementation of the decision process in this task. Specifically, a representation of both magnitude and sign is consistent with the use

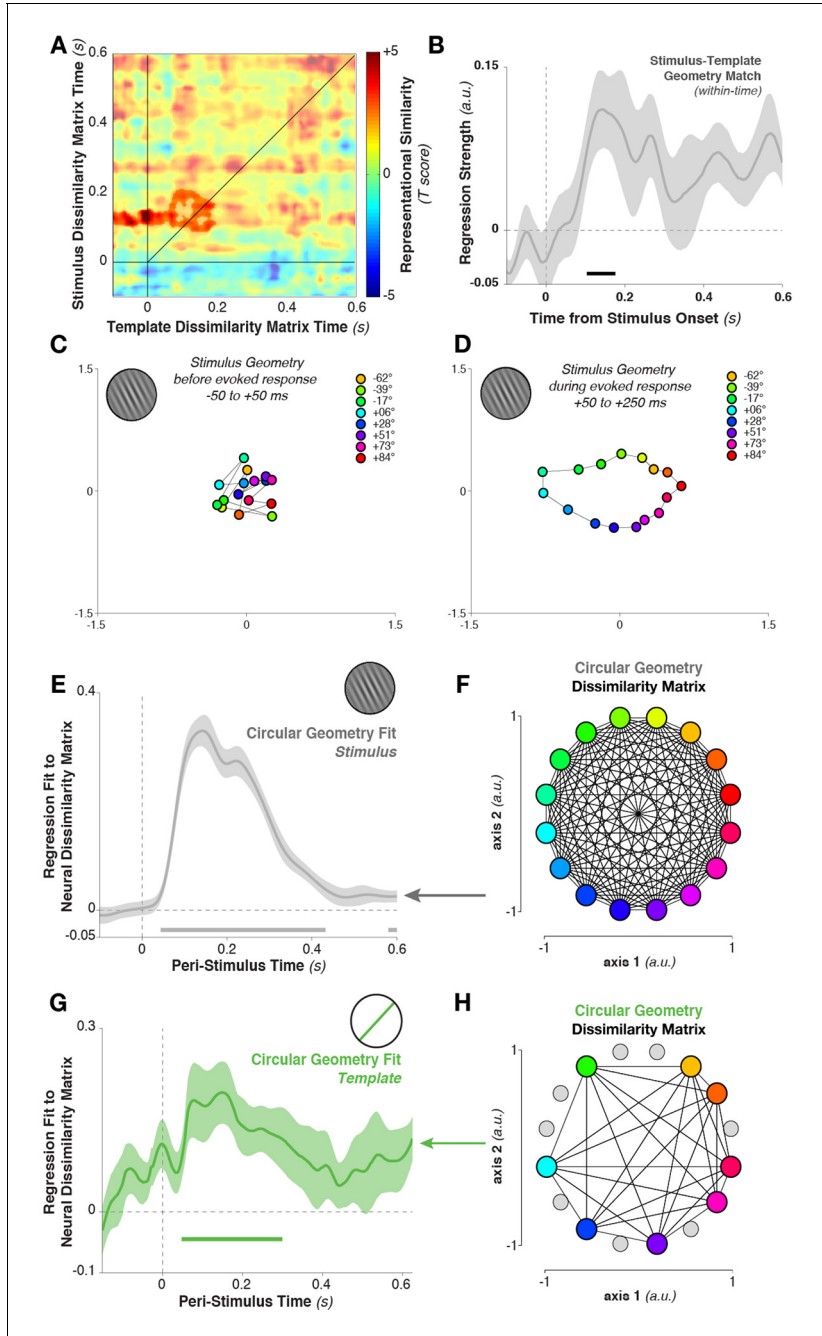

**Figure 7.** Geometry of stimulus and template coding. (**A**) The representational similarity structures between template- and stimulus-ordered responses were significantly correlated in the early stimulus-processing window (saturated colors indicate significant cluster). (**B**) The within-time comparison also showed a significant correlation in the representational similarity structure from 104 to 176 ms. Values correspond to the mean regression coefficient across all observers. Shading is between-subjects standard error of the mean. (**C**) Multi-dimensional scaling of the distances between stimulus orientations was not visible before stimulus onset. (**D**) Shortly after stimulus onset, the circular structure indicated that responses used a circular geometry. (**E** To quantify the representational structure over time, we fit (using regression) to the neural distance matrix between all angles (16 different angles, split randomly into two sets of trials, resulting in a 32×32 distance matrix of Mahalanobis distances) the distance matrix of a 16-point circular simplex, shown in (**F**). (**G**) Similarly, relationships between the eight template orientations fit a circular structure, particularly around stimulus onset time. (**H**) An example of a simplex from one session, with the eight chosen template angles highlighted in color, and the eight stimulus orientations which were never targets shown in gray.

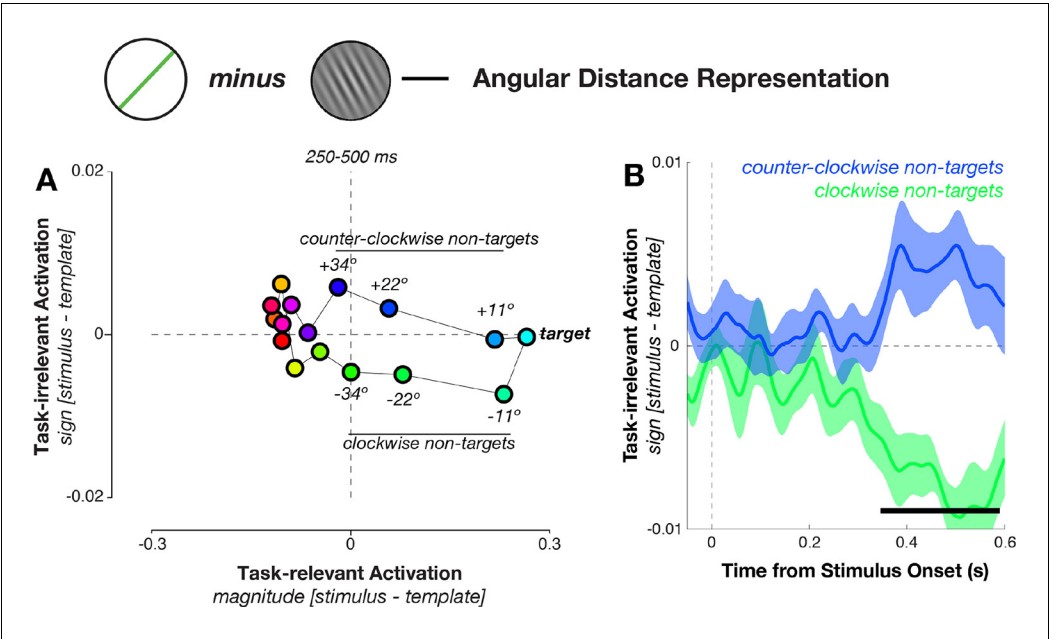

**Figure 8.** Geometry of response-related coding. (**A**) Dissimilarity structure of angular distances. Data dimensionality was reduced using PCA, and weights calculated between sensor activity and different task variables using independent training data. Mean responses for each angular distance, calculated using the left-out data, were then projected via the calculated weights onto the task axes (the magnitude and sign of the angular distance). Since the task relevance of a particular angular distance was defined solely by its magnitude, projections onto the sign axis measured sensitivity to task-irrelevant signed differences between conditions. Prior to decision onset (250–500 ms after stimulus onset), the neural geometry is elliptical: in addition to conditions separating along the target-relevant magnitude axis (horizontal), near non-targets separate along the task-irrelevant sign axis (vertical). (**B**) Task-irrelevant coding emerges approximately 350 ms after stimulus onset. Time courses for the three nearest non-targets (11°, 22°, 34° offset from target angle) separate along the task-irrelevant axis, depending on whether they are clockwise or counterclockwise to the target.

The following figure supplements are available for figure 8:

**Figure supplement 1.** Multidimensional Scaling and Pairwise Mahalanobis distances between Angular Distances.

**Figure supplement 2.** Figure is identical to Panel 8c, but includes in the graph the fit to the distance matrix provided by the linear decision value (i.e., the unsigned target proximity, stimulus—target ).

**Figure supplement 3.** Neural Population Model.

---

of a probabilistic population code (*Ma et al., 2006*; *Beck et al., 2008*). Probabilistic population codes assume that the brain uses the activation pattern across a population of neurons, each tuned to a different stimulus value (angular distance, in our case), to encode a probability distribution across the entire stimulus space (*Ma et al., 2006*; *Zemel et al., 1998*). The peak of activity of this distribution lies in or near neurons tuned to the presented angular distance. Therefore, angular distances with equal magnitude but different sign can be naturally separated in the population response, even if that dimension of the neural pattern is task-irrelevant. Importantly, this would not be the case if the entire population simply encoded the magnitude of the absolute distance (i.e., the overall match between stimulus and template, as is the case with some accumulator models of decision-making). Therefore, the presence of signed difference signals in the MEG response suggests that the brain uses a (probabilistic) population code to represent the decision variable in this task.

For illustration, we created a simple neural architecture to elaborate this argument for a population code and to link it to the neural data. The model consisted of three interconnected layers (see Materials and methods and *Figure 8—figure supplement 3* for details of model behavior). Each layer encoded information about one of the three task variables (stimulus, template, and angular distance). Each unit in a layer was tuned to a different orientation. Tuning in the decision layer represented *decision-relevant* angular distances, meaning that angles closer to 0° represented stimuli

closer to the current template. We created the same dissimilarity matrices used in the MEG/EEG analyses from synthetic responses generated by each layer in the model. Identical to *Figure 8A*, conditions with equal magnitude but different sign led to separable population responses in the template and angular distance layers. In contrast, a simpler model with only a single accumulator unit in the decision layer showed only a differentiation of conditions if they differed in magnitude, reflecting that here the signed angular difference was not encoded.

The population coding model used here is almost identical in architecture and behavior to a more elaborate biophysical model that was recently developed to predict the learning of new categories in a population of lateral intraparietal neurons in monkeys performing an orientation discrimination task (*Engel et al., 2015*; *Freedman and Assad, 2006*).

## Discussion

In a visual match-to-template orientation task, we found distinct, dynamically evolving neural responses that reflected the orientation of the stimulus and the template, as well as the angular distance between the two (i.e., the task-relevant variable). Contrary to standard models of top-down attention, we did not find a tonic activation of the template neural pattern. Instead, the template pattern emerged transiently around the time of the stimulus onset and then quickly returned to baseline.

While pattern analysis is a well-established methodology for intracranial multi-unit recordings and for fMRI, it is becoming clear that it can provide a useful approach to MEG and EEG as well. Although MEG/EEG measures neural activity at a larger scale relative to micro-electrode recordings, recent modelling demonstrates that the electromagnetic signal contains rich spatiotemporal information suitable for multivariate decoding (*Cichy et al., 2015*). Even subtle differences in dipole position or angle elicit statistically separable patterns at the scalp surface. For orientation decoding, these differences presumably depend on idiosyncrasies in the distribution of orientation columns along the cortical surface (*Cichy et al., 2015*). Such subtleties average out at the group level (or are lost during source localization due to inherent ambiguities with the inverse solution), but can be characterized within individual participants using pattern analysis (see *Stokes et al., 2015*).

This logic extends in time: small differences in the spatial distribution of activity patterns at different time points would result in idiosyncratic changes in the dipole, resulting in a time-varying signal at the scalp surface. Indeed, the cross-temporal analyses suggest that orientation-specific patterns are also time-specific (see also *Wolff et al., 2015*). In animal models, similar spatiotemporal patterns have been attributed to a cascade of neural engagement within the same brain area (*Harvey et al., 2012*) or time-specific changes in cell preferences (*Sigala et al., 2008*; *Stokes et al., 2013*). It is important to appreciate that decodability does not necessarily imply that the brain is making use of the decodable information (*Tong and Pratte, 2012*). Nonetheless, neural circuits with complex spatiotemporal dynamics could plausibly provide a rich source of information for guiding flexible (*Miller and Fusi, 2013*) and context-dependent behavior (*Buonomano and Maass, 2009*).

The rapidly changing patterns encoding the stimulus orientation raised the question of when the stimulus is compared to the template. Even though template decoding lasted up to 300 ms after stimulus onset, the template-specific neural patterns cross-generalized to stimulus-specific patterns only in the earliest encoding phase. The transient, rather than sustained, activation of template-specific patterns may reflect the reactivation of a latent code (*Mongillo et al., 2008*; *Buonomano and Maass, 2009*) that was laid down in altered synaptic weights, but which is reactivated via top-down or stimulus-driven input. Template decoding began shortly before stimulus onset, suggesting that the semi-regular timing of events may have allowed for top-down re-activation of the template (as in 'rhythmic sampling', c.f. *Schroeder and Lakatos, 2009*; *Lakatos et al., 2013*). Since template decoding peaked during the stimulus presentation period, bottom-up stimulus drive may have additionally activated the template pattern.

The rapid dynamics of stimulus decoding further raise the question of how the brain compares dynamically evolving population codes (*Laurent, 2002*; *Meyers et al., 2008*; *Stokes et al., 2013*; *King and Dehaene, 2014*). Representational similarity analysis (*Kriegeskorte and Kievit, 2013*; *Haxby et al., 2014*; *Nili et al., 2014*) permits higher-order comparisons between different task variables, even if their underlying neural patterns are different. We speculate that matched filters provide a natural solution to the problem of comparing a dynamically evolving stimulus-encoding

pattern to a template-encoding pattern: the stimulus pattern is filtered by a population that is matched to the visual characteristics of the template, leading to output that quantifies their overlap.

Unexpectedly, the task-irrelevant sign of the angular distance was encoded in the MEG/EEG response pattern. This finding could provide an interesting insight into the potential mechanism underlying perceptual decision-making in our task. Specifically, probabilistic population codes may underlie the representation of the angular distance, and encode the sign of the angular distance as a by-product of decision-making. This could be a simple result of the use of a matched filter at an earlier stage. There is evidence that stimulus orientation is represented via population codes in early visual cortex (*Graf et al., 2011*; *Berens et al., 2012*), with the activation profile across neurons tuned to many different orientations reflecting a probability distribution peaking at the orientation that is most likely present in the environment. If this population activity pattern is passed through a filter tuned to the orientation of the template, the resulting output population pattern could again reflect a probability distribution peaking at the most likely *relative* orientation of the stimulus with respect to the template. Because of the orientation symmetry of the filter mechanism, the output would also reflect the direction of the angular distance (clockwise or counterclockwise).

Recent computational and empirical work on the maintenance of items in working memory has argued that mnemonic information (such as a visual template) can be stored through a reconfiguration of synaptic weights (*Mongillo et al., 2008*; *Lewis-Peacock et al., 2012*; *Stokes et al., 2013*), without requiring strong persistent activity (*Watanabe and Funahashi, 2014*). The encoding of decision-relevant mnemonic templates in the weights of a network has the crucial advantage that processing of any new information can immediately pass through the modified weights and produce a matched filter response (*Mongillo et al., 2008*). Conceived in this way, contents in working memory are decision rules that enforce the current stimulus-response mapping (*Stokes, 2015*). Read-out at the time of the probe then consists in a perceptual decision (*Martínez-García et al., 2011*; *Pearson et al., 2014*). These hidden states, reflecting the current task context, could be in operation in our task, and would map onto the representation of the target orientation in the template layer of our toy model.

The transient reactivation of the template shortly before stimulus onset could also reflect the nature of our task, where the majority of stimuli were non-targets that may have discouraged the use of tonic template activation. For instance, single-cell studies have shown that visual distractors presented in a memory delay can disrupt tonic activity of cells coding the remembered item (in IT, *Miller et al., 1993*, and transiently in PFC, *Miller et al., 1996*). By contrast, tasks without intervening distractors may be more conducive to the use of tonic activation (*Chelazzi et al., 1993*, *1998*).

## Materials and methods

### Participants

Ten healthy, right-handed volunteers (age range: 21–27 years, 6 females) took part in the study, and were paid £10/hr for their time. Visual acuity was normal or corrected to normal. Ethical approval for methods and procedures was obtained from the Central University Research Ethics Committee of the University of Oxford. Each participant completed two experimental sessions, approximately 1 week apart, with each lasting approximately 2 hr (of which approximately 1 hr was spent on performing the task). Each participant completed a large number of trials (7680 across two sessions), providing robust within-participant statistical power for within-participant decoding.

### Experimental setup

Participants completed the MEG/EEG scan inside a sound-attenuated, dimly lit, and magnetically shielded room. Stimuli were displayed onto a rear-projection screen (placed at a viewing distance of 90 cm) via a projector (Panasonic DLP Projector, PT-D7700E) with a spatial resolution of 1024 × 768 pixels and a refresh rate of 60 Hz. Stimuli were presented using Psychophysics Toolbox (*Brainard, 1997*), running on MATLAB (Mathworks, Natick, WA). Participants responded using an optic-fibre response box by lifting their right index finger to indicate whenever they had seen a target. Participants were instructed to respond as quickly and accurately as possible.

## Task

The task required the detection of visual targets within a stream of sequentially presented stimuli. The stream consisted of oriented Gabor patches (diameter: 4° visual angle, spatial frequency: 2 cycles/°), presented foveally for 100 ms, at an average rate of 650 ms (inter-stimulus interval, ranging from 516 to 783 ms). Orientations were drawn without replacement from a set of 16 possible angles. Stimuli were equally spaced from 5.625° to 174.375°, in steps of 11.25°. The task consisted of eight brief (approximately 6 min) blocks, in which 480 stimuli were presented (resulting in a total of 3840 stimulus presentations per session). Each block began with the presentation of a target orientation (drawn at random, without replacement, from the 16 stimulus orientations), displayed centrally as a green line (4° length). Thus, each session contained eight randomly drawn target orientations (they did not need to repeat across experimental sessions). The participants were instructed to respond whenever a Gabor patch with a matching orientation appeared. Since stimuli were drawn equiprobably from the 16 possible orientations, 1/16 of all stimuli were targets. Each block was cut into three shorter segments, giving participants brief rest periods. During the rest periods, the target orientation was presented again as a reminder.

## Data sharing

In accordance with the principles of open evaluation in science (*Walther and van den Bosch, 2012*), all data and fully annotated analysis scripts from this study are publicly available at http://datashare-drive.blogspot.co.uk/2015/08/testing-sensory-evidence-against.html (see also *Myers et al., 2015*). We also hope these will provide a valuable resource for future re-use by other researchers. In line with the Organisation for Economic Cooperation and Development (OECD) Principles and Guidelines for Access to Research Data from Public Funding (*Pilat and Fukasaku, 2007*), we have made every effort to provide all necessary task/condition information within a self-contained format to maximise the re-use potential of our data. We also provide fully annotated analysis scripts that were used in this paper. Any further queries can be addressed to the corresponding author.

## Behavioral data analysis

Because of the rapid succession of stimuli, it is difficult to attribute unequivocally each response to a single stimulus. Therefore, a stimulus-response assignment procedure was designed in order to attribute, in a probabilistic fashion, each response to a single stimulus.

First, response-time (RT) distributions to stimuli were computed on the basis of their absolute angular distance (tilt) from the target orientation (from 0 to $\pm$ 90°). When RTs were averaged relative to the orientation of the stimuli, it was clear that the responses fell within a certain time window (from approximately 200 to 1000 ms), consistent with the approximately periodic presentation of stimuli. Tilt-dependent RT distributions were used to estimate the tuning of responses to the target. At each RT, the response tuning profile—the probability of a response given the tilt of the stimulus, from 0 to 90°—was fitted with a von Mises distribution having two free parameters: the peak of the distribution $P_{MAX}$, and the concentration parameter kappa $\kappa$. The von Mises distribution was constrained to be centred at the target orientation (tilt = 0), and the definition of $\kappa$ was modified such that $\kappa = 0$ indicates no tuning, $\kappa > 0$ indicates a preferred tuning for the target orientation, and $\kappa < 0$ indicates a preferred tuning for the orientation perpendicular/opposite to the target. For each subject, the tuning concentration showed a clear positive response following stimulus onset (approximately 200 to 1000 ms post-stimulus).

This tuning information was then used to assign probabilistically each response to an individual stimulus. First, for each response, all stimuli that fell into the time window during which the tuning concentration was positive were preselected. Next, among these candidate stimuli (which had different tilts with respect to the target), the stimulus that maximised the probability of a response at the observed RT was selected. The resultant RT distributions truncated the low and high RT values leaving the central part of the original RT distributions

## MEG and EEG data acquisition

Each participant completed two sessions: one MEG-only session, and one session in which EEG data were recorded concurrently. Participants were seated in the MEG scanner in a magnetically shielded room. Their legs were placed on leg rests and arms on their lap to avoid movements. Both

experimental sessions lasted approximately one hour. Participants were instructed to maintain fixation on the centre of the screen during the stimulus blocks and minimize blinking.

Neuromagnetic data were acquired using a whole-head VectorView system (204 planar gradiometers, 102 magnetometers, Elekta Neuromag Oy, Helsinki, Finland). Magnetoencephalographic signals were sampled at a rate of 1,000 Hz and on-line band-pass filtered between 0.03 and 300 Hz. The participant's head position inside the scanner was localised and tracked continuously using head-position index coils placed at four distributed points on the head. Electrodes were placed above and below the right eye for the vertical electro-oculogram (EOG) and to the side of each eye for the horizontal EOG. In addition, eye movements were monitored using a remote infrared eye-tracker (SR research, EyeLink 1000, sampling one eye at 1000 Hz, controlled via Psychophysics Toolbox, *Cornelissen et al., 2002*).

EEG data were collected in half of the sessions (for each participant), using 60 channels distributed across the scalp via the international 10–10 positioning system (*AEEGS, 1991*). Filtering, downsampling, epoching, and rejection of artefactual trials were performed on EEG data in the same way as on the MEG data. EEG data were added to all decoding analyses for the MEG+EEG sessions (except for the topographies in *Figure 3*). We found no substantial differences in decoding between MEG-only and MEG+EEG sessions, apart from a small increase in decoding sensitivity in the latter. Therefore, all within-session analyses were averaged to arrive at participant-level results.

## MEG data preprocessing

Data were preprocessed using the in-house OHBA software library (OSL), drawing on SPM8 (http://www.fil.ion.ucl.ac.uk/spm), Fieldtrip (*Oostenveld et al., 2011*), and Elekta software. The raw MEG data were visually inspected to remove and interpolate any channels with excessive noise, and were further de-noised and motion-corrected using Maxfilter Signal Space Separation (*Taulu et al., 2004*). Next, data were downsampled to 500 Hz. Remaining epochs with unsystematic noise corruption were then excluded via visual inspection. Systematic artefacts, arising from eye blinks and heart beats, were identified via independent component analysis, and regressed out of the raw data. The cleaned data were then epoched with respect to each stimulus onset (from –1 to + 1 s). In a final step, data were imported into Fieldtrip and inspected using the semi-automatic rejection tool to eliminate any remaining trials with excessive variance. All data were then baseline-corrected by subtracting the mean signal between –150 and –50 ms relative to stimulus onset (for analyses relating to the template, we used an earlier baseline, from –200 to –150 ms relative to stimulus onset, to explore the possibility that template information might be 'pre-activated' around the expected onset time. Using the standard baseline from –150 to –50 ms, however, did not change the results presented here). In addition, the data were smoothed with a 32-ms Gaussian kernel for template-based analyses to reduce noise.

## Orientation decoding

We used a population tuning curve model to recover information about the stimulus orientation from the full M/EEG signal. Instead of looking to relate imaging data to different stimulus orientations directly, each stimulus orientation is represented using weights from a linear basis set of population tuning curves. Tuning curve models are well suited to recovering information about parametric features like orientations (*Saproo and Serences, 2010*; *Brouwer and Heeger, 2011*; *Serences and Saproo, 2012*; *Garcia et al., 2013*) or colors (*Brouwer and Heeger, 2009*).

To recover stimulus orientations, data were separated into a training set (all trials from 7 of 8 blocks) and a test set (the left-out block). For all trials in the training set, we then created a matrix of 16 regressors, with the height of the each regressor on any trial reflecting that trial's stimulus orientation (i.e. a regressor was set to 1 when the corresponding orientation was presented on that trial, and to 0 otherwise). The regressor matrix was then convolved with a half-cosine basis set (raised to the 15[th] power, see *Brouwer and Heeger, 2009*), in order to pool information across similar orientations. Orientation sensitivity at each MEG/EEG sensor was then calculated by regressing the design matrix against the signal (across all 306 sensors or all 366 sensors in MEG+EEG sessions), separately for all time points in the epoch (in 4 ms steps, using a sliding window of 20 ms). We solved the linear regression equation:

$$B_1 = WC_1; \qquad (1)$$

where $C_1$ is the design matrix (16 regressors × no. of training trials), $B_1$ is the training data set (306/366 sensors × no. of training trials), and W is the weight matrix (306/366 sensors × 16 orientation values) that we want to estimate. This was done using ordinary least squares:

$$W = B_1 C_1^T (C_1 C_1^T)^{-1}; \qquad (2)$$

Overall differences in signal magnitude between sensors were modeled out using a constant regressor in $C_1$. We used W to estimate the population orientation response (or tuning curve) in the test set, $B_2$ (306/366 sensors × no. of test trials):

$$C_2 = (W^T W)^{-1} W^T B_2; \qquad (3)$$

where $C_2$ is the tuning curve, W is the weight matrix, $W^T$ is its transpose, and $W^{-1}$ is its pseudo-inverse. Since both the design matrix and the estimated weight matrix were of full rank, this approach was equivalent to using the pseudoinverse for estimation. For each trial, this curve was then zero-centered relative to the presented orientation. This procedure was repeated for each time point in the epoch before moving to the next iteration in the leave-one-out procedure. Zero-centered orientation curves were then averaged across trials.

The time course of the tuning curve was then converted into a stimulus information time course by calculating the linear slope of the tuning curve from –90° to 0°. We first averaged stimulus channels that were equidistant from 0° (i.e. +11.25° and –11.25°, +33.75° and –33.75°, etc.) and smoothed each resulting (sign-invariant) orientation channel time course (with a 16-ms Gaussian kernel). We then fit a linear slope across the orientation channels (from –90° to 0°), separately for each time point, session, and participant. Decoding accuracy was then evaluated using one-sample t-tests (against 0), under the assumption that slopes are randomly distributed around 0 if there is no stimulus information in the signal. Multiple comparisons across time were corrected for using cluster-based permutation testing (10,000 permutations, *Maris and Oostenveld, 2007*).

We used a similar approach to test for encoding of the current template orientation, with the exception that here we used a 32-ms sliding window to increase sensitivity to a more slowly evolving effect. Since there were only eight template orientations per session, and these were randomly selected from the 16 possible stimulus orientations, they were not always equally spaced across the circle. We estimated orientation tuning curves across the eight irregularly spaced angles (using eight equally spaced regressors), and then linearly interpolated the estimated tuning values at the eight intermediate values. After interpolation, the template orientation tuning curves were treated as above to derive decoding time courses.

Finally, we also applied this approach to calculating tuning profiles for information about the angular distance between the orientation of the current stimulus and the template (ranging from 0° for template matches, in steps of 11.25°, to 90°, for stimuli that were orthogonal to the template).

Onset latencies between stimulus, template, and angular distance were compared using a jack-knife approach (*Miller, Patterson and Ulrich, 1998*). We compared the onset times of significant coding (p<0.05, corrected) using t-tests. To estimate the variance of each onset time, we used an iterative procedure that left out one participant in turn and calculated the onset time of significant coding across all remaining participants. The standard error of the latency difference was calculated using a revised measure that takes into account the reduced variability caused by the jack-knife procedure (*Miller et al., 1998*). The latency difference calculated across the entire set of participants was divided by this standard error estimate to provide t-statistics that were then evaluated using the conventional t-distribution.

## Univariate orientation sensitivity analysis

In addition to the pattern analyses that averaged signals over all sensors, we tested the orientation sensitivity of individual MEG/EEG sensors, to generate a topographical distribution of the sensitivity to the three task variables (stimulus orientation, template orientation, and decision-relevant angular distances). The baseline-corrected signal at each sensor and time point in the epoch was fit (across all trials) using a general linear model (GLM) consisting of pairs of regressors containing the sine and cosine of the three task orientations, along with a constant regressor. From the pair of regression

coefficients for the sine ($\beta_{SIN}$) and cosine ($\beta_{COS}$) of an orientation, we calculated orientation sensitivity A:

$$A = \sqrt{(\beta_{COS}^2 + \beta_{SIN}^2)}; \tag{4}$$

We calculated the amplitude A expected by chance alone by permuting the design matrix and repeating the amplitude analysis 1000 times. The observed (unpermuted) amplitude was ranked within the permutation distribution of amplitudes to calculate a p-value, which was transformed into a z score using the inverse of the cumulative Gaussian distribution (with center 0 and standard deviation 1). Sensitivity at the group level was then estimated by averaging z-scored amplitudes across session, participant, and the magnetometer and two gradiometers at each sensor location. These values were then plotted as topographies to illustrate the distribution of orientation sensitivity for the three task variables.

### Cross-temporal orientation decoding

To assess the temporal stability of stimulus-specific topographies, we trained the population tuning-curve model on one time point in the epoch, and applied the estimated weights to all time points in the test data (using a sliding window of width 20 ms, applied every 12 ms). This was then repeated for all time points, creating a two-dimensional matrix of cross-temporal tuning-curve slopes (with no additional smoothing). Dynamic coding can be inferred by comparing the decoding slopes on within-time training (i.e., training and testing on time $t_1$, or time $t_2$) with the decoding slopes on between-time training (i.e., training on $t_1$ and testing on $t_2$). Our criterion for a dynamic epoch was: for each pair of time points $t_{i,j}$, coding is dynamic if the tuning curve slope is significantly higher (as measured by a paired t-test across 10 participants) within time than across time ($t_{i,i} > t_{i,j}$ AND $t_{j,j} > t_{i,j}$). Time windows of significant decoding ($t_{i,j} > 0$) and windows of significant *dynamic coding* were identified using 2-dimensional cluster-based permutation testing (i.e., across both time axes).

### Cross-generalization between stimulus and template patterns

To test whether stimulus-specific patterns cross-generalize to template-specific patterns, we repeated the cross-temporal tuning-curve analysis, but calculated weights based on the presented template orientations in the training set, and then zero-centered the tuning curves of the test set with respect to the *stimulus* orientations. Here, a significantly positive tuning curve slope at time pair $t_{i,j}$ indicates that stimulus coding around time point i shares orientation-specific topographic patterns with template coding around time point j. For consistency with the other analyses, we treated the training data as in the analyses evaluating template coding, and treated the training data as in the stimulus decoding analyses. Therefore we used a baseline of –200 to –150 ms for the training data, and smoothed with a Gaussian kernel of width 32 ms. For the test data, we used a baseline of –150 to –50 ms and did not smooth. For calculating weights, we used a sliding window of 32 ms, moving in 12-ms steps. The results were smoothed with a 20-ms Gaussian kernel. Again, we used permutation testing to correct for multiple comparisons.

### Representational similarity analysis

In light of the rapid dynamics of the population tuning curve data, we reasoned that, while the exact neural pattern might differ between time points and task variables, the information represented (as measured by their representational geometry) might be more constant over time. We tested for this possibility with RSA. Specifically, our approach involved calculating neural dissimilarity matrices between the MEG/EEG topographies evoked by different stimulus orientations. For each session, we sorted all trials by the presented stimulus orientation (into 16 bins), and then split each of these in half (separating odd and even trials). The odd–even split allowed us to compare dissimilarity structures in two independent data sets, and to verify the reliability of the RSA. For each of the 32 bins, we calculated the baseline-corrected average evoked response (across trials) at all sensors and time points. Next, for each time point (moving in 4-ms steps), we calculated the neural dissimilarity matrix by computing all pairwise Mahalanobis distances between orientations (using the within-condition covariance, pooled over all conditions). We interpret these dissimilarity matrices as reflections of the representational structure at each time point in the epoch.

In the first instance, we were simply interested in whether the neural dissimilarity structure was more stable over time than the underlying neural patterns (that were calculated in the tuning curve analyses). To test for this, we correlated (with Pearson correlations) the dissimilarity matrix from one-half of trials at one time point with the dissimilarity matrix from the other half of trials at all time points, generating a cross-temporal matrix of correlations between dissimilarity structures. If the dissimilarity structure is stable over time, this should result in significant correlations (measured via one-sample t-tests at the group level on the Fisher-transformed correlation coefficients) between time points (e.g., off-diagonal coding). We repeated this analysis for the decision-relevant angular distance.

For the analyses comparing the neural dissimilarity structures for template coding and stimulus coding, we used one-half of trials to calculate the 8×8 template-based dissimilarity matrix (on data baselined at –200 to –150 ms, as above), and the other half of trials (on data baselined at –150 to –50 ms, as above) to calculate the 8×8 stimulus-based dissimilarity matrix (using the eight stimulus orientations that also served as target orientations in that session). As above, the resulting within-time correlations were then smoothed with a 20-ms Gaussian kernel.

Next, we asked whether the neural dissimilarity structure, or geometry, was related to the parametric dissimilarity structure of the stimuli: since a 45° angle is more similar to a 60° angle than a 90° angle, the corresponding MEG/EEG topographies might be more similar as well. The stimulus dissimilarity matrix based on the pairwise angular distances between all presented orientations was regressed against the MEG/EEG dissimilarity matrix using a general linear model, fitting the model separately for each time point, session, and participant. Significant fits were assessed via one-sample t-tests. As an illustration of the presence of circular structure in the representational geometry, we projected the 32×32 dissimilarity matrix into two dimensions using multi-dimensional scaling (MDS).

We repeated geometric analyses on the dissimilarity structure with respect to the template orientation, and the decision-relevant angular distance. For the latter (angular distances), the MDS results indicated that the circular geometry of stimulus relationships was distorted by the decision likelihood. Therefore we used multiple regression to account for its possible influence (pattern component modeling, *Diedrichsen et al., 2011*). A first nuisance regressor captured the differences in the absolute decision value, i.e. the distance between the *unsigned* angular distances (i.e. the distance between 0° and –22.5° was 22.5°, but the distance between –22.5° and +22.5° was 0°, rather than 45°). The second nuisance regressor was based on the similarity in response likelihood, which we estimated by calculating the participant-wise differences in response frequency between all decision-relevant angular distances. This regressor reflected differences in response likelihood between orientations, accounting for any effect of motor preparation. This regressor accounted for the effect of linear decision value on the MEG pattern. This pattern was regressed out because it could reflect two possible decision mechanisms: population-based coding, as proposed here, or linear evidence accumulation (*Gold and Shadlen, 2007*). While the latter may still be at play in this task, we were specifically interested in dissociating the two coding mechanisms.

## Angular distance analysis

A final analysis examined how the entire population of MEG/EEG sensors dynamically encodes different task variables relating to angular distance representation. This was done by representing population responses as trajectories in neural state space (with each dimension representing a unique task variable). One approach, emulated here, has recently been described for populations of neural spike trains (*Mante et al., 2013*). First, in order to de-noise the data, we smoothed data with a 20-ms Gaussian kernel and reduced the dimensionality of the MEG signal from 306 sensors (or 306+60 sensors for MEG+EEG sessions) to 30 principal components (PCs) by calculating coefficients over the average time series at each sensor. We then fit the task variables to the reduced-dimensionality data using a GLM. The regressors were derived from the three main task variables: stimulus orientation, template orientation, and angular distance. Since all three are circular variables, we used pairs of regressors, consisting of the sine and cosine of each task angle, yielding a design matrix consisting of six regressors in total.

The fitting was done in a leave-one-block-out procedure: in turn, we held out all trials from one task block as a test set, and fit the GLM on the trials in the remaining seven blocks (the training set). The GLM was solved on normalized data (by subtracting the mean and dividing by the variance across all trials in the training set). This yielded a set of six regression coefficients ('betas') for each

time point in the trial and for each of the 30 PCs, which were then symmetrically orthogonalized (*Colclough et al., 2015*; following *Mante et al., 2013*). After normalizing the data from the test set (using the mean and variance from the training set), we calculated mean responses for all 16 angular distances in the test set (yielding a 16 angles × 30 PCs matrix). The means (16×30) were then projected onto the task axes by multiplying them, time point by time point, with the betas (30 PCs × 6 regressors) from the training set, creating a 16×6 matrix at each timepoint for each left-out block. We then averaged projections across the eight cross-validation folds. The resulting projections estimate the sensitivity of each condition (i.e., the 16 angular distances) to each task variable (i.e., the six regressors), separately for each time point in the trial.

In line with Mante and colleagues, we interpreted consistent deviations from 0 (as measured by one-sample t-tests across observers), in either direction along an axis, as task variable sensitivity. In particular, the two regressors for the angular distance partialled out task-relevant and task-irrelevant aspects of the angular distance between stimulus and template: the cosine regressor, with a maximum of 1 at 0° (targets), a minimum of −1 at the farthest non-targets ( ± 90°, in our 180° orientation space), and equal magnitudes for equivalent non-targets (e.g., 0.92 for both +11° and −11°), measured only the task-relevant aspect of the angle (i.e., the decision value, as shown in *Figure 8—figure supplement 2*). By contrast, the sine regressor is insensitive to decision value (since, in 180° orientation space, sin(0°) = sin( ± 90°) = 0), but distinguished between signed differences between non-targets (e.g., sin(+11°) = 0.38 = −sin(−11°)).

## Neural population model

To summarize our MEG results, and to illustrate how they could arise from a very simple decision circuit, we created a population-based neural coding model capable of performing the template-matching task used in our experiment. The model consisted of a three-layer architecture, with each layer consisting of neurons coding for different task variables (a stimulus layer, a template layer, and a decision layer). The stimulus layer consisted of a set of 100 units, each tuned to a different veridical stimulus orientation, with tuning determined by a von Mises distribution:

$$R_i(\theta) = exp(\kappa * \cos(\theta - \theta_i))/A; \tag{5}$$

$R_i(\theta)$ indicates the response $R$ of model unit $i$ (tuned to $\theta_i$) to orientation $\theta$, with concentration parameter $\kappa$ determining the tuning width of the response, and A reflecting a normalizing constant. Activation in the layer was then normalized to a range between 0 and 1. In the stimulus layer, the concentration parameter was set to 5 (see *Beck et al., 2008*, for the same parameter choice).

The template layer was identical to the stimulus layer, with the exception that template tuning was broader ($\kappa$ = 2), under the assumption that remembered stimuli would be encoded with lower precision than currently visible stimuli. Finally, the decision layer was identical to the template layer, with the conceptual difference that here, units were not tuned to veridical stimulus orientations, but to decision-relative orientations. In other words, activation of units tuned to near 0° in the decision layer reflected choice-relevant signals, irrespective of the current template orientation.

The stimulus and template layers were connected via a one-to-one mapping between identically tuned units (weight matrix $\mathbf{W^{ST}}$, with all connections between non-identical orientations set to 0). On each trial, the stimulus layer was initialized by setting the population response vector $\mathbf{R^S}$ in accordance with the stimulus orientation, and the template layer response vector $\mathbf{R^T}$ in accordance with the template orientation. In a second step, corresponding to a later processing stage, activation in the template layer was updated as a function of activation in the stimulus layer, by computing the element-wise product between $\mathbf{R^S}$ and $\mathbf{R^T}$. This step is similar to a Bayesian update, in which the prior distribution (the template layer response) is multiplied with the current evidence (the stimulus layer response) to produce a posterior distribution.

The crucial mapping for the task was between the template and decision layers, which consisted of an all-to-all reciprocal weight matrix $\mathbf{W^{TD}}$. The template layer unit tuned to the current target orientation had the strongest connection with the 0° unit in the decision layer (and neighboring template layer units were connected to correspondingly shifted units in the decision layer). All other connection weights fell off according to a von Mises distribution with $\kappa$ = 5 (although the exact tuning width did not substantially alter model behavior). This weight matrix shifted the response profile $\mathbf{R^T}$ in the template layer (which was still in veridical orientation space) to a response $\mathbf{R^D}$ in decision space. The decision layer response therefore permitted a direct mapping to decision- or motor-

related output regions (which are omitted here). Importantly, only the weight matrix $\mathbf{W^{TD}}$ needs to change in response to a change in the current template orientation.

Since we were mainly interested in the effects of reading out population activity in the decision layer, this model contained the simplification that codes in the three layers did not change over time. However, the dynamics of coding were not of interest for the question of whether population activity, *in principle*, could account for our neural results.

### Behavior of neural population model

The model behavior (*Figure 8—figure supplement 3E–H*) followed a simple trajectory over the course of a hypothetical trial. At the beginning of the trial, before current stimulus input has been processed, the template layer encodes the current template layer, via a bump in activation in template-tuned neurons. This input can be instantiated in the template layer via top-down input from the 0° unit in the decision layer (although other mechanisms for activating the template are also conceivable, such as periodic reactivation, e.g. *Buzsáki and Moser, 2013*; *Eichenbaum, 2013*; *Johnson and Redish, 2007*; *Lisman and Jensen, 2013*; *Schroeder and Lakatos, 2009*). This activation of the template layer around the time of stimulus onset might correspond to the decoding profile for template information in the MEG data (*Figure 3C*). Next, stimulus input is represented in the stimulus layer, again via a population activity profile peaking at neurons tuned to the currently presented stimulus (again corresponding to the decoding profile, *Figure 3A*). Stimulus layer activation is then fed forward into the template layer, where activation is multiplied point-by-point with the existing activation state. This could happen, for example, if neurons in the template layer change their gain to all input depending on their proximity to the current target orientation (*Carandini and Heeger, 2012*; *McAdams and Maunsell, 1999*; *Reynolds and Heeger, 2009*; *Silver et al., 2007*; *Treue and Trujillo, 1999*). The resulting activation profile now represents the stimulus orientation, scaled by its similarity to the template—while neurons tuned to the current orientation again have high activation, the height of that activation depends on stimulus-template similarity (and the peak is shifted towards the template orientation, with the magnitude of the shift depending on the ratio of stimulus and template tuning widths). This representation of the template and the stimulus in the same population (at slightly different but overlapping timepoints) might reflect why stimulus and template geometries cross-generalize (*Figure 7G,H*).

By passing the template layer profile on to the decision layer, it is shifted into a decision-relative (i.e. stimulus-invariant) space. Here, the response exhibits two decision-relevant features. First, the closer the current stimulus is to the template, the closer the decision layer peak is to the 0° neuron. Second, as in the template layer, the height of the decision layer profile also depends on the proximity between stimulus and template, with highest activation for targets (*Figure 8—figure supplement 3B*), and the peak dropping off for increasingly distant non-targets (*Figure 8—figure supplement 3C–E*). This decreasing amplitude may explain why near-targets were more separable than definite non-targets later in the trial epoch (*Figure 7B*). Downstream read-out units could use either the population's *peak location* (i.e. how close the maximum response is to the 0° unit, as in *Beck et al., 2008*) or its *peak activation* (as in more classical decision models) to determine whether a target is present or absent.

## Acknowledgements

We would like to thank Michael Frank, Floris de Lange, and another reviewer for helpful advice on the manuscript. We are grateful to Laura Turner and Katrina Quinn for help with data collection. This study was funded by the Medical Research Council (to M.G.S.), the Wellcome Trust (A.C.N.: Senior Investigator Award (ACN) 104571/Z/14/Z, G.R., M.W.W., and N.E.M.), St. John's College, Oxford (N.E.M.), the Fyssen Foundation and the French National Resarch Agency (V.W., grants ANR-10-LABX-0087 and ANR—10-IDEX-0001-02), an MRC UK MEG Partnership Grant (MR/K005464/1), and the National Institute for Health Research Oxford Biomedical Research Centre Programme based at the Oxford University Hospitals Trust, Oxford University. The views expressed are those of the authors and not necessarily those of the NHS, the NIHR, or the Department of Health.

## Additional information

### Funding

| Funder | Grant reference number | Author |
|---|---|---|
| Wellcome Trust | Graduate Student Fellowship (CQRTDY0) | Nicholas Edward Myers |
| Medical Research Council | HQRWVLO | Mark G Stokes |
| Fondation Fyssen | Post-doctoral research grant | Valentin Wyart |
| St. John's College, University of Oxford | Research Centre grant | Nicholas Edward Myers Mark G Stokes |
| National Institute for Health Research | Biomedical Research Centre Programme Award | Anna Christina Nobre Mark G Stokes |
| Wellcome Trust | Senior Investigator Award, 104571/Z/14/Z | Anna Christina Nobre |
| Medical Research Council | MEG Partnership Grant, MR/K005464/1 | Anna Christina Nobre |

The funders had no role in study design, data collection and interpretation, or the decision to submit the work for publication.

### Author contributions

NEM, MGS, Conception and design, Analysis and interpretation of data, Drafting or revising the article, Contributed unpublished essential data or reagents; GR, VW, Conception and design, Acquisition of data, Drafting or revising the article; MWW, Drafting or revising the article, Contributed unpublished essential data or reagents; ACN, Conception and design, Analysis and interpretation of data, Drafting or revising the article

### Ethics

Human subjects: Ethical approval for methods and procedures was obtained from the Central University Research Ethics Committee of the University of Oxford. All participants provided written, informed consent.

## Additional files

### Major datasets

The following datasets were generated:

| Author(s) | Year | Dataset title | Dataset URL | Database, license, and accessibility information |
|---|---|---|---|---|
| Myers N, Rohenkohl G, Wyart V, Woolrich M, Nobre A, Stokes M | 2015 | Data from: Testing sensory evidence against mnemonic templates | http://datadryad.org/review?doi=doi:10.5061/dryad.m57sd | Available at Dryad Digital Repository under a CC0 Public Domain Dedication |

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
