## [Decision Letter]

Thank you for submitting your work entitled "Testing sensory evidence against mnemonic templates" for peer review at *eLife*. Your submission has been favorably evaluated by Eve Marder (Senior Editor), Michael J Frank (Reviewing Editor), and two reviewers, one of whom, Floris de Lange, has agreed to reveal his identity.

Overall, we are enthusiastic about this paper. The conceptual issue at hand is topical, the experiment is cleverly designed, and the analyses are novel and insightful. However there are a number of interpretative and analysis limitations that need to be addressed for further consideration.

The reviewers have discussed the reviews with one another and the Reviewing editor has drafted this decision to help you prepare a revised submission.

Summary:

The authors measured EEG/MEG responses during a perceptual decision-making task, in which human subjects had to compare visual stimuli against a mnemonic template. They find evidence of stimulus coding, template coding, as well as coding of the signed difference between stimulus and template. Template coding appeared shortly before/during initial stages of stimulus processing, showing that the search template is transiently re-activated just prior to and during encoding of each stimulus. The authors mimic the geometry of neural responses with a toy computational model that consists of a perceptual, template and decision layer in which the decision layer uses population coding.

Essential revisions (contributed by all reviewers):

1) Reviewers agreed that while the findings are informative and novel, they are somewhat oversold, with language that is and should be interpreted more conservatively. The main advances of the paper are in showing the efficacy of MEEG methods for decoding and in demonstrating that the search template is transiently re-activated, which is worthwhile. Many of the other interpretations seem to be based on overly stringent (and probably incorrect) assumptions about how a neural code must be perfectly stationary over time, in conjunction with a possible misunderstanding about how different dipole orientations would produce different topographies without reflecting different neural computations. The paper should be reframed. The authors should consider which of the elements they have direct evidence for and which are speculations.

2) The core of the paper is in the first 4 or 5 figures; the remaining ones and analysis were less well motivated. One reviewer notes that, after Figure 3, no data are shown and, by Figure 7, interpretations are based on parameters far abstracted that it is difficult to know how those relate back to the data. As such, the manuscript is quite dense, with many highly complex analyses and results. These analyses are highly sophisticated, but it is not always clear why a particular analysis is performed. In the same vein, some details of the response-related coding analysis were also unclear. For example, why does the cosine of 'stim-template' reflect the task-relevant information, and the sine the irrelevant information?

3) Concerning the section "Stimulus and task codes vary dynamically throughout the epoch", the authors have not really shown this to be the case. We would need to see how the codes differ, for example with qualitatively changing topographies. If the neural source is oscillatory, then rapid changes in the phases would limit temporal extendability to one cycle and could lead to the misinterpretation that "multiple stimulus-specific codes must have been activated in sequence". Thus if the codes really vary over time, the topographies would have to be different in a way that is inconsistent with rotating or oscillating dipoles. This is a critical point, because the authors rely on the interpretation of the "code" being dynamic through the rest of the manuscript.

But this is more than just methodological. There is an implicit assumption here, which is that a "neural code" is an instantaneous spatial pattern, and so if it changes slightly from millisecond to millisecond, the "code" must be different. This assumption is inconsistent with neurophysiology: It is well established that neurons are sensitive to their history of inputs, and that their precise spike timing carries information about input patterns. Thus a specific representation can include time (frequency multiplexing is a simple illustration of this). Therefore, what the authors call a dynamic code could simply be a static code that has time as a dimension. If the authors were to interpret the findings more conservatively this wouldn't be problematic.

4) Rather strong statements are made about sensory representations on the basis of whole-head MEG recordings. For example, they state: "Since coding did not cross-generalize over time, multiple stimulus-specific codes must have been activated in sequence".

As noted above, we are a bit worried about the logic that the (in)ability to read out orientation signals in a generalizable fashion over time must mean that the neural codes are not stable. There are plenty of other reasons why a stable sensory code would nevertheless not generalize over time. For example, MEG measures a mixture of activity of many sources at any moment in time. If a stable source would be accompanied by a variety of other neural sources that change over time (which is almost certainly the case) this could also result in a (partial) break-down of generalization. More generally, the fact that a machine learning algorithm can "decode" a particular feature/stimulus, does not imply that the brain has "encoded" this feature. The 'decoder' could have picked up on any feature that has a non-zero correlation with the feature under investigation. Again it would be prudent to better qualify what can and cannot be concluded from the data.

5) Differences between decoding and representational geometry:

The differences in generalizability between decoding and representational similarity are intriguing. At first, it appeared that these may simply be an artifact of the method, as opposed to conceptually different characteristics of the neural activity patterns. This is better explained in the Discussion, but the reason for doing these two types of analyses, and how they provide an answer to distinct questions, could be better motivated upfront.

6) Discussion on predictive coding:

We found the link made to predictive coding somewhat of a stretch. The authors state that the signed difference between template and stimulus is passed down, in line with predictive coding. But also the stimulus itself is encoded, see Figure 3. And the prediction error in predictive coding would serve to update perception, not a decision about whether a percept is different from an internal template. This signed difference is mandated by the task in this case. Finally, the statement "Our results, using non-invasive imaging, suggest that template and stimulus coding may be similarly segregated across cortical layers" is quite over-stated. The authors investigate whole-head MEG responses, localizing their signals to roughly the back of the head. How that would suggest segregation across cortical layers is very unclear at best.

7) Equations 1 and 2: CC^T^ and W^T^W are invertible only if C and W are full column rank. Was this evaluated (for W; I assume the columns of C are all linearly independent) or was the psuedo inverse used? MEEG data are often reduced-rank, particularly after ICA cleaning. A minor aside: It might be useful to mention that these equations are the solutions to AX = b, which would help the linear-algebra-challenged readers understand the analyses.

[Editors' note: further revisions were requested prior to acceptance, as described below.]

Thank you for resubmitting your work entitled "Testing sensory evidence against mnemonic templates" for further consideration at *eLife*. Your revised article has been favorably evaluated by Eve Marder (Senior Editor), Reviewing Editor Michael Frank, and two reviewers. The manuscript has been improved but there are some remaining issues that need to be addressed before acceptance, as outlined below.

As you will see, the second reviewer is satisfied with your revised manuscript, but Reviewer 1 has some lingering concerns. Partly this reflects a difference in taste and style, but please address the specific interpretative and statistical issues that the referee raises regarding Figure 3, Figure 4, and 6. It is important that these issues are transparent.

*Reviewer #1:*

The authors made some adjustments but the overall manuscript is more or less how it was in the first submission.

The Discussion is really long, and most of it is just a rehash of the results rather than actual discussion.

The figures are still difficult to interpret, and neither the legends nor the text is very helpful. Examples:

I don't understand the outlines in Figure 4. The legend suggests it's the significance of the difference between the diagonal and off-diagonal, but the outline includes dark red regions very close to the diagonal, and it doesn't include much of the plot that should also be different from the diagonal. And what is the difference between the lines and the opaque shading in panel B?

I also don't understand Figure 6. The small black outline in panel A doesn't seem to match the darker areas in the figure, and is nothing statistically significant in panel B?

Is Figure 3 supposed to be the diagonals of Figure 4? The significances don't seem to map on to each other.

*Reviewer #2:*

I think the authors have dealt with all the reviewers' comments in an exemplary way, and I congratulate the authors with their interesting manuscript.

---

## [Author Response]

*Essential revisions (contributed by all reviewers):*

1) Reviewers agreed that while the findings are informative and novel, they are somewhat oversold, with language that is and should be interpreted more conservatively. The main advances of the paper are in showing the efficacy of MEEG methods for decoding and in demonstrating that the search template is transiently re-activated, which is worthwhile. Many of the other interpretations seem to be based on overly stringent (and probably incorrect) assumptions about how a neural code must be perfectly stationary over time, in conjunction with a possible misunderstanding about how different dipole orientations would produce different topographies without reflecting different neural computations. The paper should be reframed. The authors should consider which of the elements they have direct evidence for and which are speculations.

We thank the reviewers for the positive appraisal. We have now reconsidered our use of language throughout the paper. As a result, we have toned down our language where appropriate and clarified points that lean more on speculation informed from other studies than the results of the current study. We have also added relevant theoretical discussion to address some of the issues raised by the reviewers. These are detailed below.

We appreciate the point raised by the reviewers that neural coding is unlikely mediated by stationary activity patterns. High-temporal resolution methods provide a unique window into such dynamics, where fMRI has previously emphasised a more static patterns of activity associated with neural coding. With MEG in particular, even very subtle differences in dipole orientation can generate different patterns at the scalp surface. This property of MEG probably underpins our ability to decode stimulus orientation based on spatial patterns (see Cichy et al., 2015, NeuroImage; and relevant commentary Stokes, Wolff and Spaak, in press, Trends in Cog Sci). Formal modelling in Cichy et al. shows that even neighbouring sources can generate separable topographies within individual subjects, and this rich source of spatial information can be exploited by multivariate decoding methods. Moreover, our results suggest that these dipoles change over time, resulting in dynamic topographies that render decoding time-specific (despite stable representational structure, see RSA analyses). Such activity dynamics, evident in high-temporal resolution methods, clearly show that we need to move away from overly rigid interpretations of stable codes. In parallel, insights gained from multi-electrode recordings in animal models are helping to unravel the computational significance of dynamic coding.

We now provide further explanation in the Discussion of the possible neurophysiological basis of dynamic coding observed in MEG, and how this relates to micro-electrode recordings. We believe this is an exciting area of study at the moment, and are confident that our paper will make a positive contribution to this unfolding story.

*2) The core of the paper is in the first 4 or 5 figures; the remaining ones and analysis were less well motivated. One reviewer notes that, after Figure 3, no data are shown and, by Figure 7, interpretations are based on parameters far abstracted that it is difficult to know how those relate back to the data. As such, the manuscript is quite dense, with many highly complex analyses and results. These analyses are highly sophisticated, but it is not always clear why a particular analysis is performed. In the same vein, some details of the response-related coding analysis were also unclear. For example, why does the cosine of 'stim-template' reflect the task-relevant information, and the sine the irrelevant information?*

The second half of the Results section shows progressively more involved analyses, and we thank the reviewers for pointing out that they seemed less well motivated. We have addressed this by removing (or moving into the supplemental materials) any results that were essentially control analyses, and not directly related to advancing the main conclusions. For all results, we have rewritten sections to motivate what new aspects of the data they revealed, and why this was important. As mentioned in response to point 3, we have added a more thorough description of the representational similarity analyses in Figure 6 and Figure 7, and why they are an important complement to the time-limited decoding found in Figure 3.

The figure illustrating the neural model (Figure 8—figure supplement 3) is now in the supplemental materials, and the description of the model has been cut down to its essential point. We hope that by making this section more concise, the main argument comes out more clearly, with all the further details still available to the interested reader.

Finally, we have made an effort to explain better the analyses of response-related coding in Figure 8. The significance of this result is that it shows a neural effect at the decision stage that is not strictly necessary for performing the task, but that provides evidence for the underlying neural mechanism. This basic result is illustrated in Figure 8. Figure 8 showed a control analysis that is now in the supplemental materials.

In Figure 8, the two task axes measuring sensitivity to the response-related angle (i.e. the angular difference between current stimulus and template) were previously labelled ‘cosine’ and ‘sine’: we now define them as capturing the ‘magnitude’ and ‘sign’ of the response-related angle, respectively. We argue that only the magnitude axis is relevant to the task (and to behaviour). Our reasoning is as follows: participants have a noisy estimate of the template orientation, and a noisy estimate of the current stimulus orientation. If the absolute difference between these two estimates falls below a decision threshold, participants decide that a target is present. Because the decision threshold is non-zero, there will be some false alarms, and because both estimates are noisy, there will sometimes be misses. This accounts for the behavioural data shown in Figure 1. Importantly, this strategy relies solely on the absolute angular difference between stimulus and template (what we now term the magnitude). In our case, the magnitude of the angular distance is captured by its cosine (with a peak at 0º, when the stimulus matches exactly the template, and a trough when the stimulus differs maximally from the template). Conversely, the sign of the angular distance (i.e., whether the stimulus is clockwise or counter clockwise to the template) should be entirely irrelevant to the task. The sign of the angular distance is captured by its sine (with a peak at +90º and a trough at -90º). We have attempted to convey this in rewriting the section (Discussion).

*3) Concerning the section "Stimulus and task codes vary dynamically throughout the epoch", the authors have not really shown this to be the case. We would need to see how the codes differ, for example with qualitatively changing topographies. If the neural source is oscillatory, then rapid changes in the phases would limit temporal extendability to one cycle and could lead to the misinterpretation that "multiple stimulus-specific codes must have been activated in sequence". Thus if the codes really vary over time, the topographies would have to be different in a way that is inconsistent with rotating or oscillating dipoles. This is a critical point, because the authors rely on the interpretation of the "code" being dynamic through the rest of the manuscript.*

*But this is more than just methodological. There is an implicit assumption here, which is that a "neural code" is an instantaneous spatial pattern, and so if it changes slightly from millisecond to millisecond, the "code" must be different. This assumption is inconsistent with neurophysiology: It is well established that neurons are sensitive to their history of inputs, and that their precise spike timing carries information about input patterns. Thus a specific representation can include time (frequency multiplexing is a simple illustration of this). Therefore, what the authors call a dynamic code could simply be a static code that has time as a dimension. If the authors were to interpret the findings more conservatively this wouldn't be problematic.*

We agree that this is a crucial point. Firstly, we completely agree that neurophysiology demonstrates that neural activity is highly dynamic, even when essentially the same information is represented. We have previously reported such dynamics in monkey prefrontal cortex ([77], Neuron) and agree that diverse neurophysiological factors could supply this inherent time-dimension to patterns of neural activity (see Buonomano and Maass, 2009; Nat Rev Neurosci; Barak and Tsodyks, 2014, Trends in Neurosci; Rabinovich, Simmons and Varona, Trends in Cog Neurosci). This is exactly the purpose of our comparison between decoding and representational geometry. Together, these two complementary approaches demonstrate the reviewers’ point: dynamic changes in patterns of activity, yet stable representational structure (e.g. “a static code with a time dimension”). This *is* our notion of dynamic coding.

We use ‘neural code’ to refer to the pattern of activity that systematically maps to specific information content. We argue that coding is dynamic if this mapping changes over time. However, we appreciate that a more descriptive term would be more prudent, and would therefore be more appropriate in many parts of the manuscript. In such cases, we now substitute ‘code’ for ‘pattern’. For example: “This result is consistent with multiple stimulus-specific patterns activated in sequence”. We are happy that this more agnostic term still reflects our main meaning, but with fewer assumptions. We have now also elaborated more clearly in the Results section of the paper the relationship between changing neural patterns and stable information and added to the Discussion further information of how this notion of dynamic coding could provide an information-rich form of coding that is well-suited for complex behaviour (e.g., [56], Neuron). We believe these ideas are consistent with the reviewers’ comments.

It is also worth noting that the cross-temporal analysis approach is sensitive to oscillatory (or reversing) structure in discriminative patterns, assuming phase-locking to the task timings (see [37]). We now make closer reference to this highly relevant review paper, and note that we observe no evidence for a diagnostic pattern of event-related periodicity. We also note that lack of phase-locking would cause a break-down of such a pattern. Nevertheless, such trial-wise phase variability would also reduce evidence of time-specificity in the cross-temporal analysis. Therefore, it is unlikely that superior decoding along the diagonal axis can be accounted for by oscillatory dynamics.

*4) Rather strong statements are made about sensory representations on the basis of whole-head MEG recordings. For example, they state: "Since coding did not cross-generalize over time, multiple stimulus-specific codes must have been activated in sequence".*

*As noted above, we are a bit worried about the logic that the (in)ability to read out orientation signals in a generalizable fashion over time must mean that the neural codes are not stable. There are plenty of other reasons why a stable sensory code would nevertheless not generalize over time. For example, MEG measures a mixture of activity of many sources at any moment in time. If a stable source would be accompanied by a variety of other neural sources that change over time (which is almost certainly the case) this could also result in a (partial) break-down of generalization. More generally, the fact that a machine learning algorithm can "decode" a particular feature/stimulus, does not imply that the brain has "encoded" this feature. The 'decoder' could have picked up on any feature that has a non-zero correlation with the feature under investigation. Again it would be prudent to better qualify what can and cannot be concluded from the data.*

We appreciate the limits of M/EEG in decoding neural states, and now take more care to highlight the caveats in our interpretation. First and foremost, we soften our previously over-strong statements (e.g., using ‘could’ rather than ‘must’); but we also now discuss in more detail how extraneous factors could limit decodability. For example, systematic non-linear interactions could in principle limit decodability across different time points. Such interactions would limit cross-temporal generalisation, which also raises the question how such dynamics are handled in the brain. We agree with the reviewers that decodability does not necessarily reflect encoding in the brain (and now touch on this point in the Discussion). Until we understand exactly how activity patterns are read out by downstream areas, we cannot be sure what information is meaningfully encoded in the signal. Therefore, we must assume a more agnostic interpretation: if information is decodable in principle, then it could be neurally relevant code.

5) Differences between decoding and representational geometry:

*The differences in generalizability between decoding and representational similarity are intriguing. At first, it appeared that these may simply be an artifact of the method, as opposed to conceptually different characteristics of the neural activity patterns. This is better explained in the Discussion, but the reason for doing these two types of analyses, and how they provide an answer to distinct questions, could be better motivated upfront.*

As mentioned above (in response to points 2 and 3), this analysis formalises some of the important issues raised by the reviewers. In brief, the representational similarity analyses show that dynamic patterns of activity can nonetheless yield time-stable information content. We have now explained this in more detail in the Results section.

6) Discussion on predictive coding:

*We found the link made to predictive coding somewhat of a stretch. The authors state that the signed difference between template and stimulus is passed down, in line with predictive coding. But also the stimulus itself is encoded, see Figure 3. And the prediction error in predictive coding would serve to update perception, not a decision about whether a percept is different from an internal template. This signed difference is mandated by the task in this case. Finally, the statement "Our results, using non-invasive imaging, suggest that template and stimulus coding may be similarly segregated across cortical layers" is quite over-stated. The authors investigate whole-head MEG responses, localizing their signals to roughly the back of the head. How that would suggest segregation across cortical layers is very unclear at best.*

The reviewers make a fair point that the link between our findings and predictive coding might be made more transparent. In the Discussion, we have now removed the comparison to prediction errors and the speculation about layer-specific responses. We do believe that our findings fit most naturally with the idea of gain modulation as articulated by Feldman and Friston. Therefore, we have left a concise reference linking our work to theirs in the text.

7) Equations 1 and 2: CC^T^ and W^T^W are invertible only if C and W are full column rank. Was this evaluated (for W; I assume the columns of C are all linearly independent) or was the psuedo inverse used? MEEG data are often reduced-rank, particularly after ICA cleaning. A minor aside: It might be useful to mention that these equations are the solutions to AX = b, which would help the linear-algebra-challenged readers understand the analyses.

Matrices C and W were indeed of full rank, but analyses were done on the pseudo-inverse of W. We have amended this oversight in the Methods section, and have added a first equation (B = WC) for clarity.

[Editors' note: further revisions were requested prior to acceptance, as described below.]

As you will see, the second reviewer is satisfied with your revised manuscript, but Reviewer 1 has some lingering concerns. Partly this reflects a difference in taste and style, but please address the specific interpretative and statistical issues that the referee raises regarding Figure 3, Figure 4, and 6. It is important that these issues are transparent.

We are grateful to the editors and reviewers for the care and time they have taken in reviewing our manuscript. We have now made an effort to articulate our results as transparently as possible. We are particularly grateful to Reviewer 1 for spotting an apparent discrepancy between Figure 3. This turned out to be the result of an oversight when updating figures to match precise analysis parameters between within-time and cross-time analyses. We have updated the relevant figures in the revised version of the manuscript. The results remain qualitatively and statistically the same, and the discrepancy is now resolved. Please see the response below for further details on this. The first and final authors have double checked all the uploaded data and analysis scripts to ensure that the methods and reported results are consistent and as easy as possible to follow.

*Reviewer #1: The authors made some adjustments but the overall manuscript is more or less how it was in the first submission. The Discussion is really long, and most of it is just a rehash of the results rather than actual discussion.*

We agree, and have now taken greater care to avoid recapping results and have focused on points for discussion.

*The figures are still difficult to interpret, and neither the legends nor the text is very helpful. Examples:*

We apologize that the text and legends still lacked clarity in places. We have made a renewed effort to explain what we show (see detailed responses below).

*I don't understand the outlines in Figure 4. The legend suggests it's the significance of the difference between the diagonal and off-diagonal, but the outline includes dark red regions very close to the diagonal, and it doesn't include much of the plot that should also be different from the diagonal. And what is the difference between the lines and the opaque shading in panel B?*

This is a crucial point, and we are grateful for an opportunity to clarify it. The color in the figure indicates the strength of decoding, with the cluster-corrected significant regions indicated by the opaque shading. Our index of dynamic coding requires both on-diagonal timepoints to be significantly higher than the corresponding cross-temporal (off-diagonal) generalization, which is why the dynamic cluster (black outline) only appears in a relatively small part of the plot where corresponding on-diagonal coding is high. Importantly, the dynamic cluster includes some points near the diagonal: while these still show significant decoding, it is nonetheless significantly lower than the corresponding on-diagonal timepoints. In this sense, we interpret the underlying pattern to have changed significantly, even though there is also some shared pattern allowing for significant cross-generalisation. Conversely, there are timepoints on the diagonal (approx. 350-400 ms, near the end of the significant cluster) that show significant decoding that is nonetheless too weak to be significantly larger than the off-diagonal – for this reason, the off-diagonal dynamic coding cluster (i.e. the black outline) might not always extend quite as far along the y-axis as the significant on-diagonal cluster.

In Figure 4 (template decoding), there was no significant drop from on-diagonal to off-diagonal decoding, so there is no black outline. As with Figure 4, the opaque shading showed the timepoints with significant decoding.

We have attempted to make these points clearer in the presentation of the results (Results section) and in the legend to Figure 4.

*I also don't understand Figure 6. The small black outline in panel A doesn't seem to match the darker areas in the figure, and is nothing statistically significant in panel B?*

We used the same conventions here as in Figure 4: significant clusters are indicated by the opaque shading (there is one large significant cluster in each panel), and a significant off-diagonal reduction (compared to on-diagonal) is shown in the black outline. There was no significant off-diagonal reduction in panel b. We apologize that the figure legend was confusing. We have now rewritten it to improve clarity.

*Is Figure 3 supposed to be the diagonals of Figure 4? The significances don't seem to map on to each other.*

Yes, these two should match. That is, we should expect the significant cluster in Figure 3 (within-time template decoding) to match significant on-diagonal timepoints in Figure 4 (cross-temporal template decoding). The discrepancy was an oversight on our part: it was caused by an accidental inclusion of an old version of the within-time analysis using slightly different baselining (from -250 to -100 ms, instead of -200 to -150 ms). We have now updated the figure accordingly, and as expected, the within-time decoding (Figure 3) is also significant just before stimulus onset (-72 ms onwards). A comparison of the old and new result shows almost identical curves, but the onset of template decoding is now significantly earlier than the onset of stimulus decoding. We have updated the corresponding part of the Results. However, this change in the onset of the significant window does not change our interpretation of the results, since we had already discussed the likelihood that part of the template reactivation could occur shortly before stimulus onset (end of Discussion) as the same pre-stimulus time period was near-significant (p < 0.07) even in the previously reported analysis.

As mentioned in the original version of the paper, we are committed to the principles of Open Science, and therefore make all data and analysis scripts freely available online for complete transparency. In preparing the final version to accompany this paper, we found other minor inconsistencies, none of which significantly change the qualitative or statistical aspects of our results. Specifically, the cross-time decoding analysis of the stimulus and relative angle (presented in Figure 4) was based on a sliding window size of 16 ms rather than the stated 20 ms. Both yield almost identical results, and therefore we have simply updated the figure with the appropriate analysis without any other change. The cross-generalization analysis between templates and stimuli (Figure 5) had also used a different baseline (as stated above). Using the more consistent baseline yields the same cluster in time, although it is only a strong trend after correcting for multiple comparisons (p = 0.06). We have updated the results accordingly. The representational similarity comparison of templates and stimuli (Figure 7) was calculated with a different baseline as stated (-250 to -100 ms, i.e., consistent with the old version of Figure 3). We have now updated this figure using the baseline reported in the Methods (-200 to -150 ms). Again, the results remain practically unchanged, however we note that the significant cluster in Figure 7 has shifted slightly. Finally, the angular distance analysis (Figure 8) had also used a slightly different baseline window. Once again, the updated results are almost indistinguishable.

We are now satisfied that all reported results faithfully reflect the methods as originally outlined.